# Connecting high-resolution 3D chromatin organization with epigenomics

Fan Feng[1], Yuan Yao[2], Xue Qing David Wang[3], Xiaotian Zhang[4] & Jie Liu[1,2✉]

The resolution of chromatin conformation capture technologies keeps increasing, and the recent nucleosome resolution chromatin contact maps allow us to explore how fine-scale 3D chromatin organization is related to epigenomic states in human cells. Using publicly available Micro-C datasets, we develop a deep learning model, CAESAR, to learn a mapping function from epigenomic features to 3D chromatin organization. The model accurately predicts fine-scale structures, such as short-range chromatin loops and stripes, that Hi-C fails to detect. With existing epigenomic datasets from ENCODE and Roadmap Epigenomics Project, we successfully impute high-resolution 3D chromatin contact maps for 91 human tissues and cell lines. In the imputed high-resolution contact maps, we identify the spatial interactions between genes and their experimentally validated regulatory elements, demonstrating CAESAR's potential in coupling transcriptional regulation with 3D chromatin organization at high resolution.

[1] Department of Computational Medicine & Bioinformatics, University of Michigan, Ann Arbor, MI, USA. [2] Department of Computer Science & Engineering, University of Michigan, Ann Arbor, MI, USA. [3] Division of Hematology, Department of Medicine, Keck School of Medicine, University of Southern California, Los Angeles, CA, USA. [4] Department of Pathology, University of Michigan, Ann Arbor, MI, USA. ✉email: drjieliu@umich.edu

                                                                1

Whereas 3D chromatin organization at the large scale of topologically associating domains (TADs) and compartments has been well-characterized in many cell and tissue types by Hi-C technology[1], our understanding of fine-scale 3D chromatin organization at the nucleosome resolution has just begun[2–4]. With the increasing evidence that fine-scale chromatin organization at the nucleosome resolution is closely related to epigenomic state[5,6], one intriguing question to ask is whether we can accurately extrapolate such high-resolution chromatin contact maps from epigenomic features such as chromatin accessibility, histone modifications, and transcription factor binding profiles. To explore this, we proposed CAESAR (Chromosomal structure And EpigenomicS AnalyzeR), a deep-learning approach to predict nucleosome-resolution 3D chromatin contact maps from existing epigenomic features and lower-resolution Hi-C contact maps.

Our model leverages cutting-edge deep-learning approaches to identify representations relevant to high-resolution chromatin organization. In particular, 1D convolutional and graph convolutional layers[7] identify epigenomic patterns over the linear chromatin fiber and over the 3D spatial chromatin organization that is relevant to impute high-resolution chromatin contact maps. With existing high-resolution Micro-C contact maps, Hi-C contact maps, and a number of cell-type matched epigenomic data on human H1-hESC (hESC), mouse ESC (mESC), and human foreskin fibroblasts (HFF), we systematically evaluated the model's performance across different chromosomes, across different cell types, and across different species. In the experiments, the model accurately imputes many fine-scale chromosomal structures that Hi-C sequencing fails to detect, including short-range chromatin loops and stripes. The model is more accurate at imputing evolutionarily conserved regions, active A compartment, and early-replicating regions, which indicates that the fine-scale 3D chromatin organization is strongly influenced by the nature of the epigenomic factors in these regions. The imputed chromatin contacts also recapitulate enhancer activities previously elucidated by CRISPRi experiments[8], and manifest expression quantitative trait loci (eQTLs) previously profiled by GTEx project[9]. CAESAR is also coupled with an attribution method that identifies epigenomic features explanatory to these fine-scale 3D chromatin structures. The explanatory features help to further subtype fine-scale chromatin structures and elucidate the interplay between histone modifications and nucleosome level chromatin organization.

CAESAR connects 3D genome organization with epigenomics at nucleosome resolution and unprecedented scale. First, compared with previous computational models for imputing Hi-C contact maps, such as HiCPlus[10], HiCGAN[11], and HiC-Reg[12], CAESAR reaches a much higher resolution. Since the majority of epigenomic activities (TF binding and histone modifications) take place at the nucleosome resolution, it is desirable to develop the predictive model that connects epigenomics and chromatin organization at the nucleosome resolution. Second, although previous models EpiTensor[13] and DeepTACT[14] also reconstruct sparse 3D chromatin interactions from epigenomics at an ultra-high-resolution, CAESAR learns from real Micro-C contact maps and predicts all chromatin contacts within a distance range, which reveals diverse fine-scale structures such as stripes, TADs, and polycomb interactions between repressive regions. Third, different from Akita[15] and DeepC[16] which predict chromatin contact maps from conserved DNA sequences, CAESAR generates tissue-specific or cell line-specific predictions from epigenomic features. Therefore, it imputes an unprecedented number of high-resolution human chromatin contact maps, including 57 tissue samples, 16 cell lines, 12 primary cells, and 6 in vitro differentiated cells. The imputed high-resolution contact maps are shared on a web server (https://nucleome.dcmb.med.umich.edu/), which allows users to easily navigate these fine-scale chromatin structures and the corresponding explanatory epigenomic features. In addition, CAESAR includes an attribution component, which reveals detailed relationships between 3D chromatin organization and epigenomic features.

## Results

**A deep-learning model imputing high-resolution chromatin contact maps.** We proposed CAESAR, a supervised deep-learning model to impute chromatin contact maps at nucleosome resolution. CAESAR's inputs include a lower-resolution Hi-C contact map and a number of histone modification features (e.g., H3K4me1, H3K4me3, H3K27ac, and H3K27me3), chromatin accessibility (e.g., ATAC-seq), and protein binding profiles (e.g., CTCF) (Supplementary Note 2). CAESAR captures the Hi-C contact map as a graph $\mathcal{G}$ with nodes representing genomic regions of 200-bp long, weighted edges representing chromatin contacts between the regions, and $N$ epigenomic features modeled as $N$-dimensional node attributes. The architecture of CAESAR (Fig. 1a, Supplementary Fig. 1, and Supplementary Note 3) includes ordinary 1D convolutional layers which extract local epigenomic patterns along the 1D chromatin fiber, and graph convolutional layers which extract spatial epigenomic patterns over the neighborhood specified by $\mathcal{G}$. The concatenated outputs from the convolutional layers capture all relevant features for one particular 200-bp bin, which are further fed into two parallel output layers—a fully connected layer predicts the contact profile for each 200-bp bin, and an inner product layer predicts loops between bins. The outputs from the fully connected layer and the inner product layer are summed up as CAESAR's final output. Using Micro-C contact maps from hESC, mESC, and HFF as the prediction target, the model was trained with backpropagation[17], in which the aforementioned convolutional features were learned adaptively. Other than leveraging a number of epigenomic features, our model architecture differs from HiCPlus[10] and DeepHiC[18] which treats Hi-C contact maps as images and performs grid-convolution to improve the resolution. With the graph convolutional networks and additional epigenomic features, CAESAR not only enhances the resolution of contact maps but also predicts the structures which are not captured by Hi-C, including polycomb repressive regions, short-range loops, and stripes (Fig. 1b).

**Accurately predicting high-resolution chromatin contact maps.** With existing Micro-C data on mESC, hESC, and HFF, we evaluated CAESAR in three different sets of experiments, including a cross-chromosome experiment, a cross-cell-type experiment, and a cross-species experiment, so as to evaluate the model's generalizability in different scenarios. In the cross-validation experiment on hESC, we divided the human chromosomes into a train set, a test set, and a tune set of similar sizes (Supplementary Notes 4 and 5). CAESAR and two baseline models, including HiCPlus[10] which only used low-resolution chromatin contact maps, and Hi-C-Reg[12] which only used epigenomic features, were trained with the train set and evaluated with the test set (Supplementary Note 6). We used the tune set to tune hyperparameters. For CAESAR and HiC-Reg, 6 epigenomic features were used, including ATAC-seq, CTCF, H3K4me1, H3K4me3, H3K27ac, and H3K27me3. CAESAR outperformed HiCPlus and HiC-Reg in terms of the stratum-adjusted correlation coefficient (SCC) with the observed Micro-C contact map (Fig. 2a). The results demonstrated that it is necessary to leverage both the contact maps and epigenomic features in the prediction of high-resolution contact maps. In the cross-cell-type experiment, we used the same train set of chromosomes to build a model on HFF, and then tested it on hESC with the same test set of chromosomes as in the cross-chromosome experiments. The

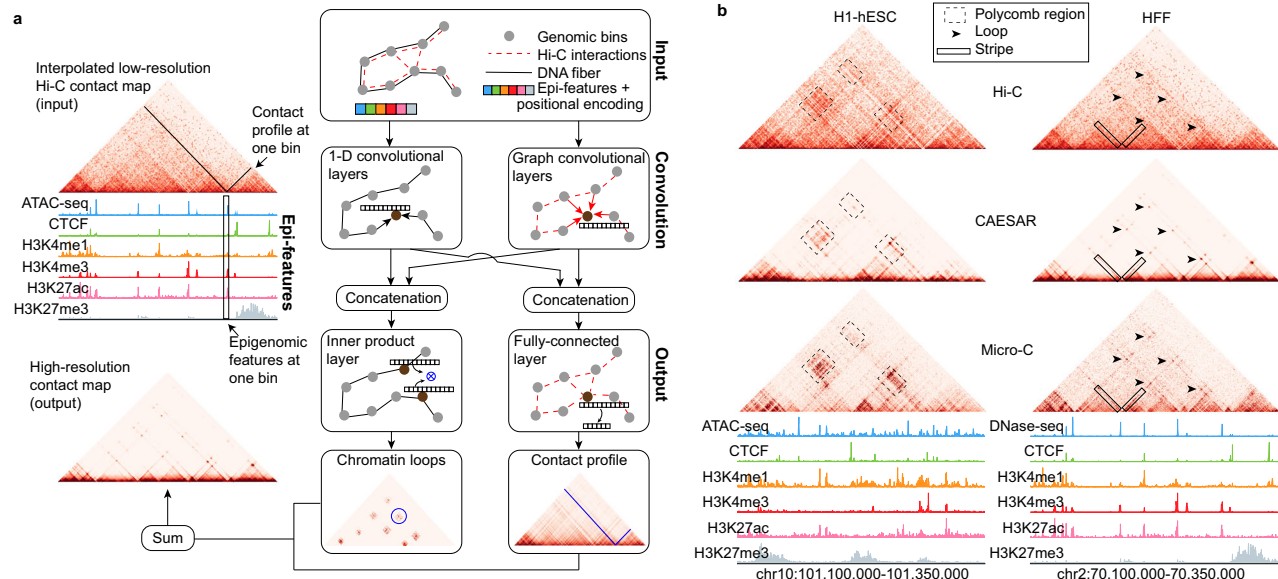

**Fig. 1 Overview of the model. a** Model architecture. The model inputs are a Hi-C contact map and a number of epigenomic features including histone modifications, chromatin accessibility, and protein binding profiles. The lower-resolution Hi-C contact map is first interpolated into a 200 bp resolution contact map, and then transformed into a graph $\mathcal{G}$ in which the nodes represent 200-bp genomic bins and the edges represent the interpolated contacts between the nodes. Positional encoding is unrelated to Hi-C or epigenomic data and only encodes node order in the genome. The epigenomic features and positional encoding are assigned to the corresponding nodes as node attributes. The inputs are fed into 1D convolutional and graph convolutional layers to generate hidden representations, which extract features from both nearby genomic regions along the 1D DNA sequence and spatially contacting regions specified by $\mathcal{G}$. The output layers take input the hidden representations and predict the contact profile at each 200-bp bin as well as the chromatin contacts between bins. **b** In an example region, the polycomb interactions are accurately predicted by CAESAR. In another example region, loops and stripes undetected by Hi-C are accurately predicted by CAESAR.

HFF-trained model imputed almost as well as the hESC-trained model for chromatin contacts within 100 kb and 200 kb range (Fig. 2b). In the cross-species experiment, we trained the model on mESC and tested the performance on hESC. In order to stay consistent with cross-chromosome and cross-cell-type evaluation, we also divided mouse chromosomes into train, tune, and test sets of similar sizes. We trained the model with mESC's train set and then tested its performance on the same aforementioned test set of hESC. It was observed that the model trained on mESC also moderately generalized to hESC, and the generalization deteriorates as the contact distance increases.

In addition, we tested CAESAR's performance in predicting fine-scale structures including loops and stripes. In the test set of HFF, CAESAR captured 72% of the loops and 63% of the stripes from Micro-C contact maps, whereas only less than 1% were captured from the input Hi-C contact maps (Fig. 2c, e, Supplementary Fig. 4, and Supplementary Notes 7, 8, and 9). Since loops called from two Hi-C replicates only agree ~60%[19], we believe that our imputed contact map recovers a good portion of these fine-scale structures. By piling up all the loop and stripe regions called from the Micro-C contact maps, we observed comparable enrichment from our predicted high-resolution contact maps and the observed Micro-C contact maps, but the pile-up results from the input Hi-C contact maps showed little enrichment (Fig. 2d, f).

Chromatin contact maps imputed by CAESAR also show comparable cell-type variability as real Micro-C contact maps in terms of SCC and cell-type-specific fine-scale structures, including chromatin loops and stripes (Supplementary Figs. 5 and 6 and Supplementary Note 10).

**Factors influencing CAESAR's performance.** In order to optimize CAESAR's efficiency, we next explored the factors

influencing its performance. As CAESAR's principle inputs are epigenomic and Hi-C data, we began by evaluating the minimum required the number of datasets to achieve good imputed results. Four sets of epigenomic features were chosen based on common availability (Fig. 3a), and we observed comparable performance among the 13-epi, 7-epi, 6-epi, and 3-epi models (Fig. 3b). Although the SCC of the 3-epi model (including ATAC-seq, CTCF, and H3K27ac) did not drop significantly, it over-predicted fine-scale structures (Supplementary Note 8). Therefore, we recommend using the commonly profiled 6 epigenomic features in CAESAR. We also asked what is the requirement for input Hi-C contact maps. Using Hi-C data from Rao et al.[1] and Krietenstein et al.[3], we tested four contact maps, including the original Hi-C contact maps with around 1 billion contacts, two down-sampled Hi-C contact maps with 100 million and 10 million contacts, and a surrogate Hi-C contact map with 1 billion contacts aggregated from four unmatched cell lines. The surrogate contact map acts as a replacement when no chromatin contact map is available for a particular cell type. Although the SCC curve does not drop significantly with the downsampled contact maps, surrogate Hi-C performs better (Fig. 3c). The model trained with surrogate Hi-C can still capture 69% of the loops and 61% of the stripes from Micro-C contact maps in the test set (Supplementary Fig. 3). Therefore, if the matched Hi-C contact map is unavailable to complement the epigenomic data in a particular analysis, a surrogate contact map can be used in CAESAR.

We further investigated the relationship between CAESAR's performance, measured with Spearman's correlation between the imputed and the observed Micro-C contact maps, and evolutionary conservation, measured with phastCons scores. It was observed that the model imputed more accurately in the regions with higher evolutionary conservation (Fig. 3d). In addition, we also discovered that the model imputes more accurately in A compartment than B compartment, and in early-replicating

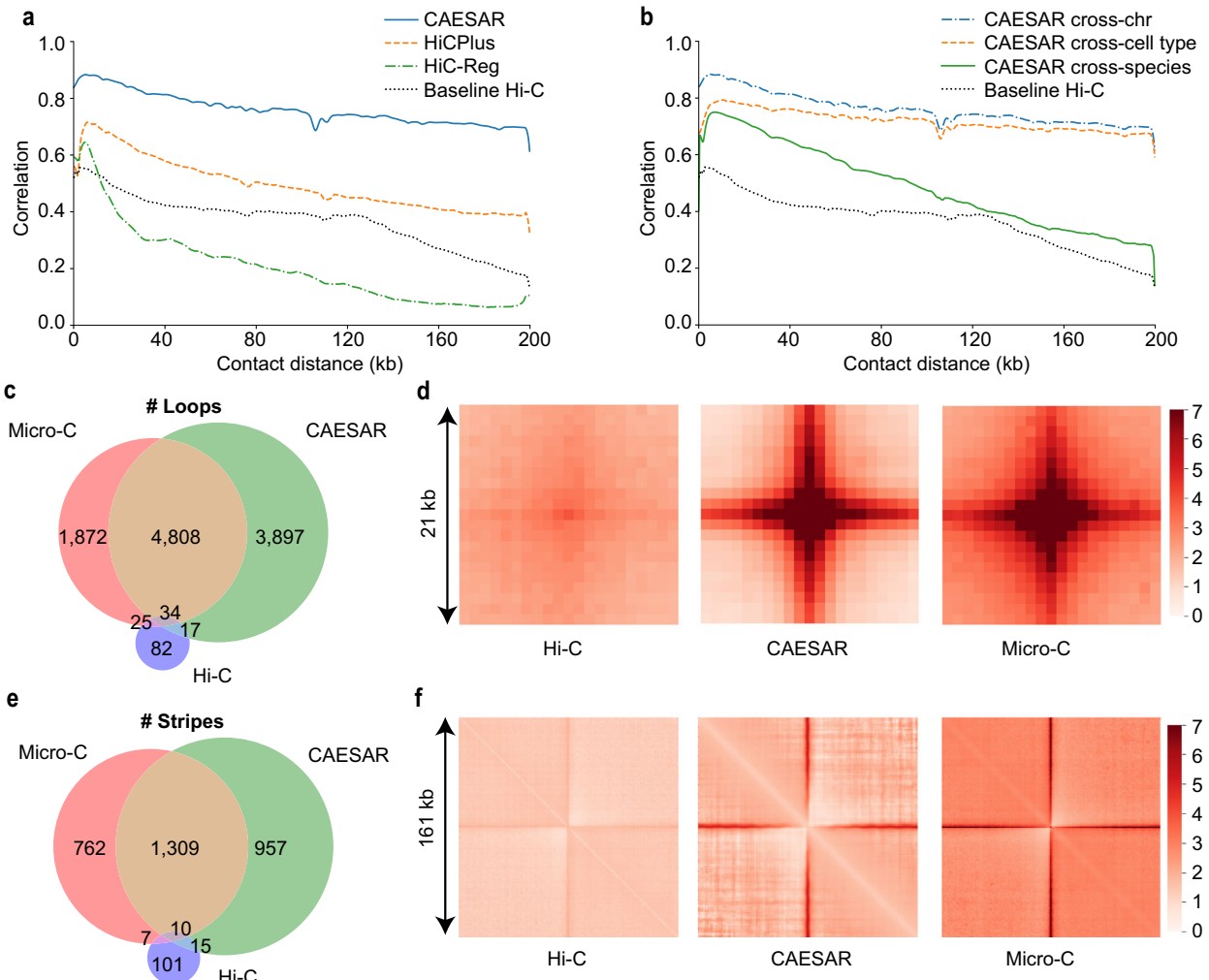

**Fig. 2 Evaluating CAESAR's performance in multiple tasks. a** The distance-stratified Pearson's correlation with the observed Micro-C contact map from CAESAR and two baselines, HiC-Reg and HiCPlus, in a cross-chromosome experiment. The black dotted lines in (**a**, **b**) are the correlation between the input Hi-C contact map and the observed Micro-C contact map. **b** The distance-stratified Pearson's correlation with the observed Micro-C contact map from CAESAR in (1) a cross-chromosome experiment (train on hESC train set and test on hESC test set), (2) a cross-cell-type experiment (train on HFF train set and test on hESC test set), and (3) a cross-species experiment (train on mESC train set and test on hESC test set). **c** The Venn diagram of the loops called from (1) the input Hi-C contact map, (2) the CAESAR-imputed contact map, and (3) the observed Micro-C contact map. **d** The pile-up visualization of the loops called from (1) the input Hi-C contact map, (2) the CAESAR-imputed contact map, and 3) the observed Micro-C contact map. **e** The Venn diagram of the stripes called from (1) the input Hi-C contact map, (2) the CAESAR-imputed contact map, and (3) the observed Micro-C contact map. **f** The pile-up visualization of the stripes called from (1) the input Hi-C contact map, (2) the CAESAR-imputed contact map, and (3) the observed Micro-C contact map.

regions than late-replicating regions (Fig. 3e, f). The results indicate that fine-scale chromatin organization is more closely related to the 6 epigenomic factors at evolutionarily conserved regions, A compartment, and early-replicating regions.

**Recapitulating CRISPRi-validated enhancer activities**. With publicly available epigenomic data, we imputed high-resolution chromatin contact maps for 15 human cancer cell lines (Supplementary Table 4b). In some cancer cell lines, noncoding regions with their regulating genes have been interrogated by CRISPR interference (CRISPRi) technology[8]. The profiled CRISPRi score indicates genomic loci's capability to regulate an essential gene, and the peaks (both positive and negative) often correspond to enhancers and promoters.

We used the CRISPRi scores profiled near two essential genes, *MYC* and *GATA1*, to validate our imputed contact maps. On the imputed contact maps for the chronic myelogenous leukemia cell K562, *MYC* gene strongly interacts with *PVT1*, which matches with the peaks of CRISPRi scores at *PVT1* locus (Fig. 4a). The imputed contact map also showed a significant interaction between *GATA1* and *HDAC6*, which matches the CRISPRi score peak at *HDAC6* locus (Fig. 4b). The matching of chromatin contacts and CRISPRi score peaks demonstrates our model recapitulates gene-enhancer interactions in cancer cell lines.

**Recovering eQTL-gene interactions**. With the large-scale epigenomic data available from ENCODE and Roadmap Epigenomics Project, we imputed the high-resolution contact maps for 57 human tissue samples and two cell lines—IMR-90 and GM12878 (Supplementary Tables 4a and b). With eQTLs profiled by GTEx[9], we asked whether our imputed chromatin contacts are enriched between genes and their eQTLs in the corresponding tissue or cell

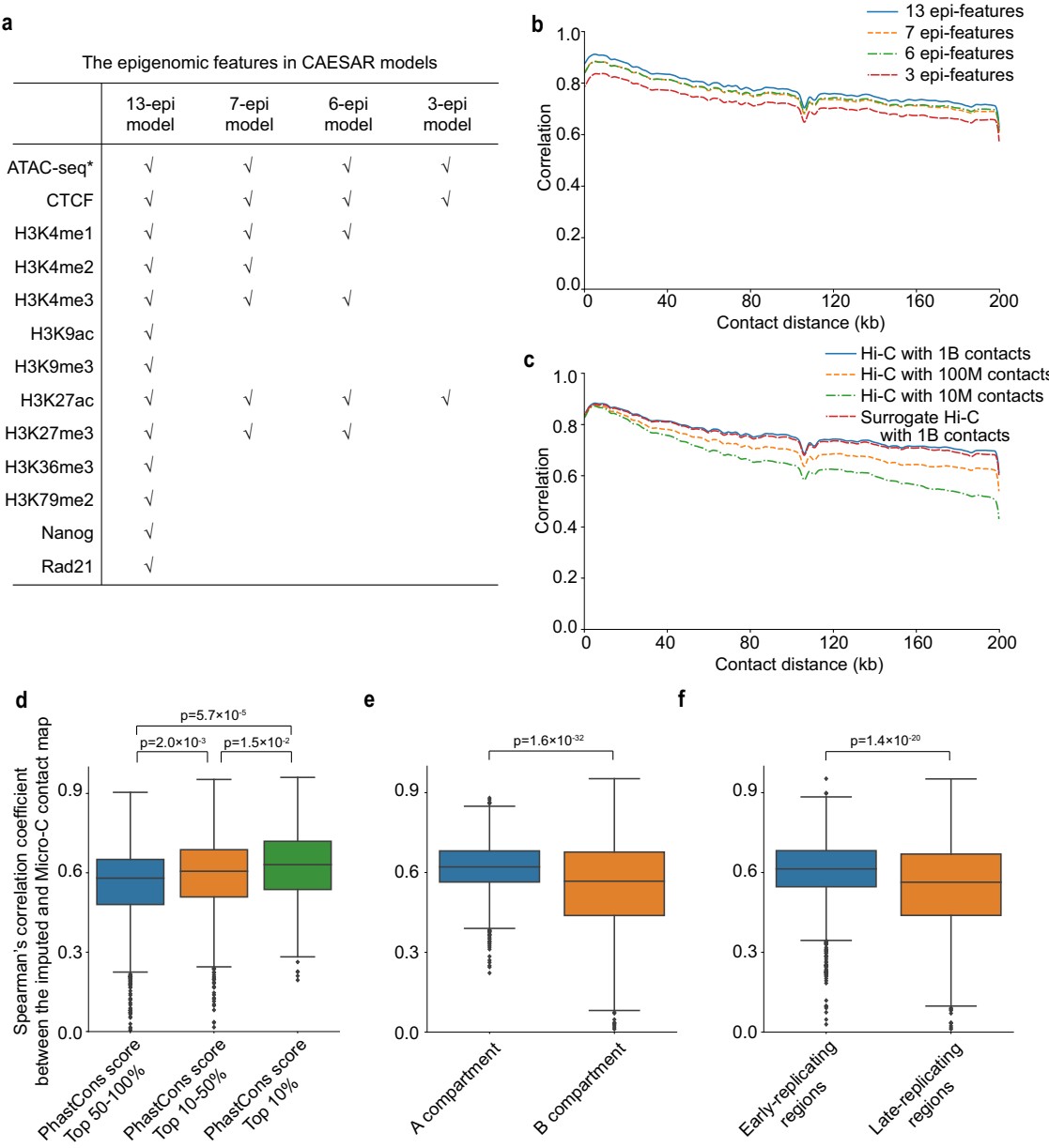

**Fig. 3 The relationships between CAESAR's performance with Hi-C quality, the number of epigenomic features, evolutionary conservation, A/B compartments, and early/late replication timing. a** The epigenomic features in 13-epi, 7-epi, 6-epi, and 3-epi CAESAR models are listed in the table, which are chosen based on common availability. **b** The distance-stratified Pearson's correlation with the observed Micro-C contact map from CAESAR in a cross-cell-type experiment with different numbers of epigenomic features (i.e., 13, 7, 6, and 3). **c** The distance-stratified Pearson's correlation with the observed Micro-C contact map from CAESAR in a cross-cell-type experiment when (1) using the original Hi-C contact map with about 1 billion contacts, (2) randomly downsampling the Hi-C contact map at different downsampling rates (resulting in 100 million and 10 million chromatin contacts), and (3) using a surrogate Hi-C contact map with 1 billion contacts aggregated from HFF, GM12878, IMR-90, and K562 with equal proportions. **d** The model performance in a specific region is quantified by Spearman's correlation coefficient between the CAESAR-imputed and the Micro-C contact map. In cross-chromosome and cross-cell-type experiments, the model performance (i.e., Spearman's correlation coefficient) is significantly correlated with evolutionary conservation evaluated by sequence alignment scores (n[regions] = 1203, 960, and 240, one-sided t test). In all the boxplots, the center line indicates median; the box limits are upper and lower quartiles; the whiskers are 1.5 × interquartile range; the points are outliers. **e** In cross-chromosome and cross-cell-type experiments, the correlation coefficient is significantly larger in A compartment than in B compartment (n[regions]=1,018 and 1,388, one-sided t test). **f** In cross-chromosome and cross-cell-type experiments, the correlation coefficient is significantly larger in early-replicating regions than in late-replicating regions (n[regions] = 1205 and 1109, one-sided t test).

line. Previous works[20] have shown eQTLs are enriched in tissue-specific frequently interacting regions on Hi-C contact maps at 40 kb resolution, but a large portion of eQTLs reside too close to their gene transcriptional start sites (TSS) to be seen on a low-resolution contact map (Fig. 5a and Supplementary Fig. 7a). For example, two eQTLs that are specific in the pancreas and lung, respectively,

both locate in chr16:57,950,000–58,050,000. The loop between the pancreas-specific eQTL and its target *USB1* gene can only be called from the CAESAR-imputed contact maps of the pancreas. The loop between lung-specific eQTL and its target *TEPP* gene can be called from the CAESAR-imputed contact maps of both lung and pancreas, which demonstrates some tissue-specific eQTLs do

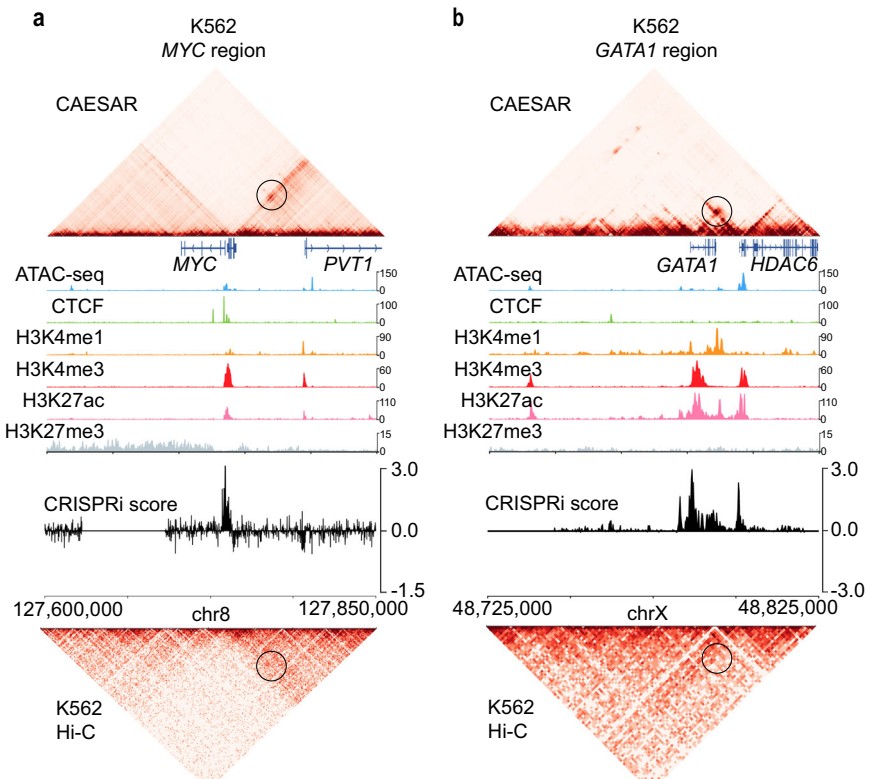

**Fig. 4 The interactions between genes and their CRISPRi-validated enhancers in CAESAR-imputed contact maps. a** The CAESAR-imputed contact map of K562 at *MYC* region (chr8: 127,600,000–127,850,000) demonstrates significant contacts between *MYC* and *PVT1*, which agree with CRISPRi score peaks, but are not shown on the original input Hi-C contact map. The magnitude of the epigenomic features is the observed value divided by the genome-wide average. **b** The CAESAR-imputed contact map of K562 at *GATA1* region (chrX: 48,725,000–48,825,000) demonstrates significant contacts between *GATA1* and *HDAC6*, which agree with CRISPRi score peaks, but are not shown on the original input Hi-C contact map.

not necessarily correspond to exclusive loops in the tissue (Fig. 5a). More examples are visualized in Supplementary Fig. 7e and f.

To evaluate the overall contact enrichment between eQTLs-TSS pairs, we piled up the contact regions between tissue-specific eQTLs and their gene TSS. In the pile-up results of twelve tissue/cell lines, seven CAESAR-imputed contact maps (adrenal gland, heart left ventricle, IMR-90, pancreas, sigmoid colon, spleen, and transverse colon) have the highest contact values for their tissues/cell line-specific eQTL-TSS interactions. Another four CAESAR-imputed contact maps (GM12878, lung, stomach, and tibial nerve) also have close-to-highest contact values for their tissues/cell line-specific eQTL-TSS interactions. These results demonstrate that tissue/cell line-specific enhancer-promoter interactions are recovered by CAESAR. In addition, the moderate enrichment on Micro-C and CAESAR-imputed contact maps from unmatched tissue/cell lines further demonstrate the eQTL-TSS interactions are not necessarily exclusive even if the eQTLs are tissue or cell line-specific (Fig. 5b). This suggests that some fine structural interactions are conserved across tissues or cell types but the regulatory functions remain specific.

**Identifying epigenomic features relevant to fine-scale 3D chromatin organization**. Although deep-learning models are often referred to as "black boxes", their outputs can be traced back and interpreted. In our model, we used integrated gradient[21] to attribute the predicted chromatin contacts to each genomic locus of each input epigenomic feature. The attribution results illustrate which parts of the epigenomic features are the most determinative for the model's predictions. By attributing the

entire contact map to all epigenomic features, we evaluated the overall contribution for each feature, and low attribution is another reason for leaving H3K4me2 out from the 7-epi model besides limited availability (Supplementary Fig. 8a).

This method can be applied to arbitrary regions on the contact map, which allows us to connect fine-scale structures with the most explanatory epigenomic features. Surprisingly, many of the peaks in the input epigenomic features do not necessarily help the model to predict fine-scale structures. For example, the H3K27ac peaks showed negative attribution in predicting the stripe in Fig. 6a and the loop in Fig. 6b (Supplementary Note 12). With attribution calculated by integrated gradient, the predicted chromatin structures can be further analyzed and subtyped (Supplementary Fig. 8e, f and Supplementary Note 11).

## Discussion

Our study connects nucleosome-resolution chromatin structures with epigenomic features. Leveraging the currently available Micro-C contact maps for hESC, mESC, and HFF from the 4DN consortium and the corresponding epigenomic profiles from ENCODE and Roadmap Epigenomics Project, we systematically mapped 1D epigenomic profiles to fine-scale 3D chromatin structures with CAESAR. The mapping was validated by high SCCs with observed Micro-C contact maps and the accurate capture of fine-scale loops and stripes. CAESAR can be applied to generate high-resolution contact maps for any cell line or tissue as long as their common epigenomic features are profiled. Our model further connects transcriptome with fine-scale structures and epigenomics by identifying the spatial interactions between genes and regulatory elements. Therefore, the imputed

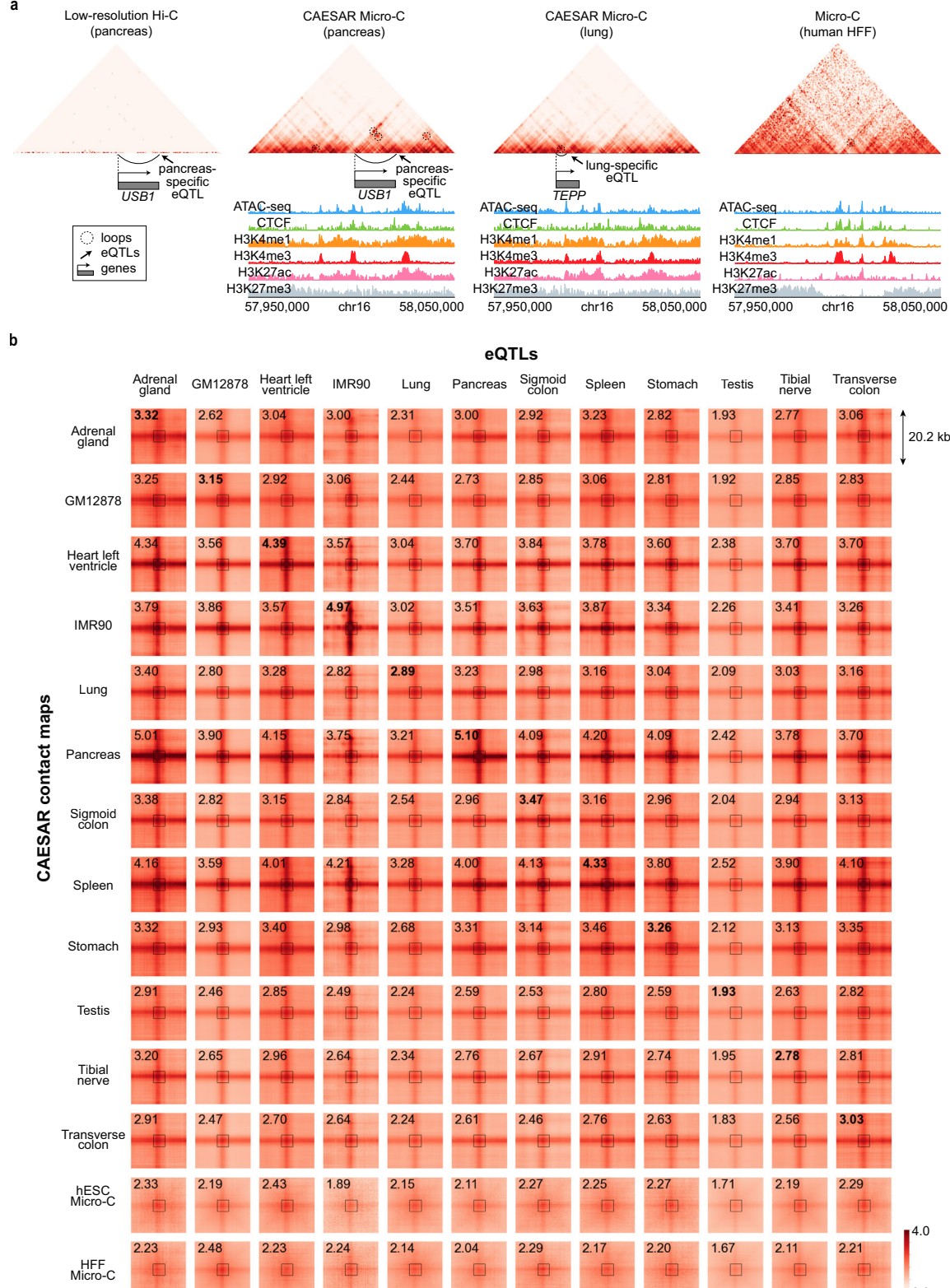

**Fig. 5 The enrichment of eQTL-gene interactions in CAESAR-imputed contact maps. a** The loop between gene *USB1*'s TSS and its pancreas-specific eQTL, which cannot be observed on the original Hi-C contact map, appears on the CAESAR-imputed contact map for the pancreas. Although gene *TEPP*'s eQTL is lung-specific, the corresponding loop can be called from the CAESAR-imputed contact maps for both lung and pancreas. **b** Pile-up analysis of the chromatin contacts between eQTLs and their corresponding gene TSS from twelve different human tissues and cell lines on CAESAR-imputed contact maps and Micro-C contact maps. The average contact values in the central 5 × 5 squares are marked on the plots, in which the bold fonts indicate that eQTLs and CAESAR-imputed contact maps are from the same tissue/cell line.

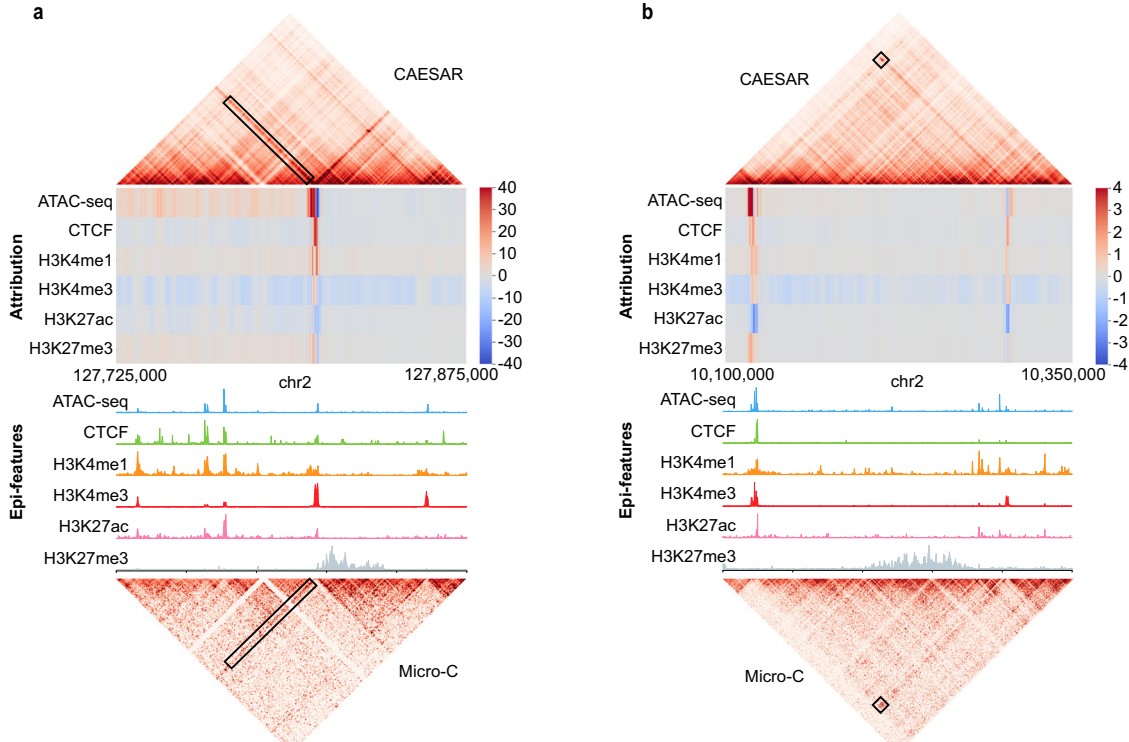

**Fig. 6 Attributing CAESAR outputs to epigenomic features via integrated gradient.** Larger attribution magnitudes indicate more contribution to the model's prediction. **a** The significant attribution of the particular stripe are from its anchor. Although all six epigenomic features have peaks at the anchor locus, the model predicts the stripe mostly from (1) ATAC-seq and CTCF peaks at the anchor, and (2) H3K4me1 modification surrounding the anchor. **b** The significant attribution of the particular loop are from its two anchors. Although H3K27ac have peaks at the left anchor locus, its contribution is negative towards predicting the loop. The CTCF binding at the anchors and H3K4me1/H3K4me3 modifications next to the anchors have positive attribution in predicting the loop.

high-resolution contact maps will be useful for target finding, hypotheses generating, and other downstream analyses. All imputed human chromatin contact maps across 57 tissues, 16 cell lines, 12 primary cells, and 6 in vitro differentiated cells have been made publicly available on our web server (http://nucleome.dcmb.med.umich.edu/) for ease of access by biomedical researchers to perform further analyses (Supplementary Table 1 and Supplementary Note 13).

While CAESAR presents a method to investigate fine details of 3D chromatin structure, we note that it is an evolving methodology with certain shortcomings that can be improved. First, since Micro-C data mostly outperforms Hi-C in the detection of short-range interactions, CAESAR also performs best at genomic distances of less than 200 kb. As a result of this, CAESAR-imputed contact maps are not well suited for analyses of large 3D chromatin structures such as compartments. Second, because Micro-C and Hi-C generate short-read sequences, our study is still limited to pairwise chromatin contacts, and therefore higher-order interactions are insufficiently studied. Third, our analyses showed that CAESAR performed well according to multiple evaluation metrics, yet there was clear bias towards A compartment, evolutionarily conserved regions, and early-replicating regions. This is likely a reflection that the epigenomic features in the study are generally more enriched in these regions. As such, it is possible that including additional epigenomic features may shift this bias effect accordingly. Fourth, though CAESAR demonstrated clear relationships between epigenomic features and 3D fine-scale chromatin organization, we did not observe significant improvement in imputed contact maps with the increasing number of epigenomic datasets. This suggests that epigenomic data may not explain all the features observed in 3D chromatin

organization. There may be unexplored layers of genetic and/or epigenetic information that play a role in the organization of chromatin inside the nucleus. So far, CAESAR demonstrated a framework for jointly analyzing 3D chromatin structures and 1D epigenomic features at a matched resolution, and further integration of 1D DNA sequences is possible. For example, our model can potentially include DNA sequences as features and elucidate 3D QTLs[22] in the context of high-resolution chromatin organization.

## Methods

**Model training.** CAESAR takes both epigenomic features and Hi-C contact maps as inputs. Based on the availability of epigenomic features, we trained four models with different epigenomic features—one model with 13-epi-features including ATAC-seq, CTCF, H3K4me1, H3K4me2, H3K4me3, H3K9ac, H3K9me3, H3K27ac, H3K27me3, H3K36me3, H3K79me2, Nanog, and Rad21; one model with seven epi-features including ATAC-seq, CTCF, H3K4me1, H3K4me2, H3K4me3, H3K27ac, and H3K27me3; one model with six epi-features including ATAC-seq, CTCF, H3K4me1, H3K4me3, H3K27ac, and H3K27me3; and one model with three epi-features including ATAC-seq, CTCF, and H3K27me3. Due to the high computational burden, it is impossible to feed the entire contact map into the memory, and therefore we used a 250-kb sliding window with 50-kb step length along the diagonal (e.g., 0–250,000; 50,000–300,000; 100,000–350,000; ...) to select the regions and fed them one by one into the model.

We split all chromosomes into train, tune, and test sets of similar sizes (Supplementary Note 4). We used the train set to train the parameters and the tune set to choose hyperparameters (Supplementary Note 5). During training, the parameters were optimized by minimizing the mean squared error (MSE) with Adam algorithm[23]. Because the model has two parts, one for predicting contact profiles and one for predicting loops (Supplementary Fig. 1 and Supplementary Note 3), we employed a sequential training strategy as follows. First, the loop predicting part was trained, in which the model was optimized targeting only the observed Micro-C contacts in loop regions (i.e, 10 kb × 10 kb squares centered at Micro-C loops) instead of the entire contact map. Second, we trained the contact profile part with the residual contact map (i.e., the observed Micro-C contact map

minus the outputs of the loop predicting part). The outputs from the two parts were summed up to generate the predicted contact maps.

**Evaluation experiments**. Three sets of cross-validation experiments were performed. First, the cross-chromosome model was trained with the train set of hESC, and tested on the test set of hESC. Second, the cross-cell-type model was trained with the train set of HFF, and tested on the test set of hESC. Third, the cross-species model was trained with the train set of mESC, and tested on the test set of hESC.

To compare CAESAR with baselines and evaluate how much they improve original Hi-C contact maps, we calculated the stratum-adjusted correlation coefficient (SCC)[24] between the observed Micro-C contact map and (1) the CAESAR-imputed contact map, (2) the contact maps imputed by other baseline methods (Supplementary Note 6), and (3) the interpolated Hi-C contact map. Other than evaluating SCC, we also called and compared the loops and stripes from the CAESAR-imputed contact maps, the Micro-C contact maps, and the Hi-C contact maps. We implemented a fast loop calling approach and a stripe calling approach to call loops and stripes at 1 kb resolution (Supplementary Notes 7 and 8 and Supplementary Fig. 2). We compared the loops and stripes called from (1) the CAESAR-imputed contact map, (2) the observed Micro-C contact map, and (3) the interpolated Hi-C contact map to generate a Venn diagram. We piled up all stripe and loop regions called from Micro-C contact maps in (1) the CAESAR-imputed contact map, (2) the observed Micro-C contact map, and (3) the interpolated Hi-C contact map.

**Correlating model performance with evolutionary conservation, A/B compartments, and replication timing**. We tested whether the model performance is correlated with evolutionary conservation, A/B compartments, and replication timing. The genome was split into 250 kb mutually exclusive fragments. For each fragment, we imputed the OE-normalized contact map at 200-bp resolution and smoothed it with a $5 \times 5$ uniform kernel. We calculated Spearman's correlation coefficient between the imputed and the observed Micro-C contact maps to evaluate the model's performance at this fragment.

The 100-way hg38 phastCons scores[25] were used to quantify evolutionary conservation. We processed the hg38 phastCons scores into 250 kb resolution and performed a correlation test between the model performance (i.e., the Spearman's correlation coefficients) and the phastCons scores. Then, the fragments were clustered into three groups, top 10%, top 10–50%, and the others, according to their phastCons score ranking. A box plot of spearman's correlation coefficients was plotted for each group.

The A/B compartments were called at 250 kb resolution. By checking the sign of the first eigenvector of the normalized contact map[26], we separated all 250-kb bins into two groups. The one with more enriched H3K27ac was labeled as A compartment, while the other B compartment. The one-sided Student's $t$ test was applied to identify whether the two groups have significantly different Spearman's coefficients.

Similarly, early-late replication timing is defined by the sign of the two-stage repli-seq signal[27]. We processed the repli-seq signal at 250 kb resolution and separated the fragments into two groups, early-replicating regions and late-replicating regions. The one-sided Student's $t$ test was applied to identify whether the two groups have significantly different Spearman's coefficient.

**Attribution by integrated gradient**. We used integrated gradient to identify each input dimension's contribution to the output. Let $\mathbf{X}$ denote the input epigenomic signals

$$\mathbf{X} = \begin{bmatrix} X_1^{(s_1)} & \dots & X_n^{(s_1)} \\ \dots & \dots & \dots \\ X_1^{(s_m)} & \dots & X_n^{(s_m)} \end{bmatrix} \in \mathcal{R}^{m \times n}, \quad (1)$$

in which $s_1, \dots, s_m$ are $m$ epigenomic signals (e.g., ATAC-seq, CTCF, etc.) and $1, 2, \dots, n$ are the indices of 200-bp bins. CAESAR takes $\mathbf{X}$ as input and learns a mapping function $F: \mathcal{R}^{m \times n} \rightarrow \mathcal{R}^{n \times n}$ to predict $n \times n$ chromatin contacts between $n$ bins (denoted as $\mathbf{Y}$). Integrated gradient[21] attributes the output to each input dimension of $\mathbf{X}$ by calculating a path integral of the gradient $\frac{\partial \mathbf{Y}}{\partial \mathbf{X}}$. Gradient $\frac{\partial \mathbf{Y}}{\partial \mathbf{X}}$ is a measure to quantify how much each dimension of $\mathbf{X}$ influences $\mathbf{Y}$, which reveals the contribution from each input dimension. The path integral starts from a predefined "background" $\mathbf{X_0}$ and ends at $\mathbf{X}$, and thus it accumulates the contributions of each input dimension from the background to real input $\mathbf{X}$[28]. Here, we used a matrix of all zeros as the epigenomic background. As demonstrated in[21], a straight-line path is efficient at disentangling the input features. Formally, the attribution of the $t$-th epigenomic signal $s_t$ at bin $i$ is:

$$A(X_i^{(s_t)}) = \int_{\alpha=0}^{1} \frac{\partial y}{\partial \gamma_i^{(s_t)}(\alpha)} \frac{\partial \gamma_i^{(s_t)}(\alpha)}{\partial \alpha} d\alpha \quad (2)$$

in which $y$ can be $\mathbf{Y}$ or a part of $\mathbf{Y}$, $\frac{\partial y}{\partial \gamma_i^{(s_t)}}$ is the gradient, $\gamma$ is the path, and $\gamma_i^{(s_t)}(\alpha)$ is the dimension corresponding to $X_i^{(s_t)}$ in the path.

By calculating the attribution toward the entire output, we obtained an overall attribution from each epigenomic feature, in which the scale of its absolute value indicates the magnitude of its importance (Supplementary Fig. 8a). Alternatively, the attribution can be calculated for an arbitrary region on the contact map, e.g., a chromatin loop or a chromatin stripe, and used for further subtyping of these loops and stripes (Supplementary Note 11 and Supplementary Fig. 8e, f).

**High-resolution contact map imputation for 91 human tissues and cell lines**. As the cross-cell-type model is validated, we used the trained model to impute high-resolution chromatin contact maps for other human tissues and cell lines. We collected the epigenomic signals from a total number of 57 tissue samples, 16 cell lines, 12 primary cells, and 6 in vitro differentiated cells (Supplementary Note 2). If the ATAC-seq signal was unavailable, DNase-seq was collected as an alternative. The 6-epi CAESAR model trained with both hESC and HFF's train set was used. For IMR-90, GM12878, and K562, we used their deeply sequenced (above 1B contacts) Hi-C contact maps as input. For cell lines or tissues without Hi-C or with only shallowly sequenced Hi-C, we used the surrogate Hi-C as input (Supplementary Note 2).

**Validation of imputed contact maps with CRISPRi in cancer cell lines**. The profiled CRISPRi score indicates the strength a genomic locus regulates a gene, and the peaks (both positive and negative) correspond to enhancers and promoters. We binned the CRISPRi scores at 200-bp resolution. On the imputed high-resolution contact maps, we selected the region near *MYC* gene (chr8: 12,765,000–12,785,000) and *GATA1* gene (chrX: 48,725,000-48,825,000) for K562. The contacts in these regions were jointly analyzed with CRISPRi scores.

**Validation of imputed contact maps with eQTLs in human tissues**. To process the raw eQTL data, we identified the 200-bp bin where each variant and its corresponding TSS locates and the contacts between the variant bin and TSS bin. We only kept the eQTL-TSS "bin pairs" which are (1) less than 180 kb apart (200 kb prediction range − 20-kb window size), and (2) specific in only one tissue or cell line. The piled-up analysis was applied to the eQTL-TSS interactions in (1) the CAESAR-imputed contact map, (2) the Micro-C contact map of hESC and HFF. For each eQTL-TSS pair, a square region (101 pixels × 101 pixels) centered at their contact was collected. The regions from each contact map were piled up, averaged, and further visualized.

**Reporting summary**. Further information on research design is available in the Nature Research Reporting Summary linked to this article.

## Data availability
The data that support this study are available from the corresponding author upon reasonable request. All data used in our model training, tuning, and evaluation are publicly available. All imputed human chromatin contact maps are publicly available on our web server (http://nucleome.dcmb.med.umich.edu/). Source data are provided with this paper (Supplementary Note 1 and Supplementary Tables 2 and 3).

## Code availability
Our figures are generated from the computational results of CAESAR. The source code, trained model, and example imputation results are shared in our GitHub repository (https://github.com/liu-bioinfo-lab/caesar). The entire imputed contact maps of tissues/cell lines are shared on our web server (https://nucleome.dcmb.med.umich.edu/).

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

## Acknowledgements

The research was supported by NIH awards R35HG011279 (J.L.) and R03OD030599 (J.L.). The authors deeply appreciate the feedback from the 4DN Joint Analysis Working Group, Drs. George Zhang and Russell Ryan from the University of Michigan.

## Author contributions

F.F. and J.L. conceived the idea. F.F. and J.L. designed the model and algorithms. F.F. implemented the model and performed the experiments. F.F. and Y.Y. collected the relevant datasets. Y.Y. implemented the webserver. F.F. and J.L. wrote the manuscript. X.Q.D.W. and X.Z. provided feedback regarding experiments and the manuscript.

## Competing interests

The authors declare no competing interests.
