## [Peer Review File · Nature Communications]

REVIEWER COMMENTS

Reviewer #1 (Remarks to the Author):

The goal of this project is to use low-resolution Hi-C data and ChIP-Seq/ATAC-Seq to predict higher-resolution chromatin interaction maps. There have been at least 4-5 publications on this topic in different journals. Therefore, I think the novelty of this work is limited. Further, there is no evidence or functional validation to test the results from their algorithm. My detailed comments are listed below:

Major:

1. Because a lot of downstream analysis in this work are based on the model trained with the surrogate Hi-C map, it would be crucial to add further validations. What are the accuracies of the loop and stripe predictions based on the model trained with surrogate Hi-C maps.
2. Resolutions used in this work are confusing. In the main text, they mentioned that they predicted the contact profile at 200bp resolution (line #68). However, based on the supplementary notes, both loops and stripes were called at 1kb.
3. It seems benchmark was not performed thoroughly. What is the performance of a model that directly predicts the entire contact map?
4. According to Fig. 3c, using the surrogate Hi-C map can achieve a comparable performance to the model using cell-specific Hi-C map with 1 billion contacts. So, what is the contribution of Hi-C in the model?
5. It has been shown that architectural stripes are frequently associated with active enhancers (H3K27ac). However, in Fig. 6 and Fig. S4, the authors showed that H3K27ac has negative attribution in predicting stripes. This is counter intuitive. Is this novel biological discovery or prediction error?
6. The APA plots throughout the paper look strange to me: the signals are enriched in the whole region (not sure if they used any normalization in these plots)? Also in Fig. 5a-5b, it seems the loops and stripes are marked arbitrarily, because many other comparably evident dots are not marked.

Minor:

1. Fig. 5a-5b, Hi-C maps (the leftmost column) should use the same resolution to show the difference.
2. Not clear which map did the authors use to identify A/B compartments in fig. 3e, Hi-C, Micro-C, or the imputed map?
3. In the method section (line 323), the authors mentioned that the pile-up analysis of the eQTL-TSS pairs was performed using Micro-C map of both hESC and HFF cell lines. However, I did not see the results for hESC Micro-C.

Reviewer #2 (Remarks to the Author):

The authors propose a supervised deep learning method that uses 1D chromatin features and low(er) resolution contact maps to predict high-resolution contact maps and their features including loops and stripes. This certainly could be a useful method however there are several important issues with the work and the manuscript as it stands.

1. Unlike claimed, CEASAR is not "the first study connecting 3D genome organization with epigenomics". Indeed, this journal has published in 2016 a method called EpiTensor that predicts Hi-C maps from 1D chromatin features. This work is not even cited! There are also several other methods such as DeepTACT (Bioinformatics, 2019), methods that predict

contact maps from CTCF motifs and ChIP-seq data (the cited HiC-Reg method is just one of them) and two recent methods in Nature Methods that use DNA sequence to predict contacts (Fudenberg et al and Schwessinger et al). In short, the paper is missing in citations of critical literature making it seem more novel than it is. I fully understand the differences between the proposed method and all these other works and it is imperative to highlight these properly.

2. In general, I find the predicted contact maps a bit artificial. The visual pattern (many stripe-like features) suggests that the method is highly biased by the 1D coverage of each region which gets reflected as the absence of depleted contact regions for some loci. This is immediately visible in Fig 1 Hi-C vs CEASAR vs Micro-C comparison. This issue will significantly impact the usefulness of the overall method for the broader community.

3. The above issue is also apparent when comparing different entities in Figure 2c. The aggregate pattern of the CEASAR loops does not mimic that of Micro-C loops. There does not seem to be a distance-dependent decay as one moves away from the center pixel.

4. The loop and stripe detection are done using simple and customized methods rather than established ways of doing these. Therefore, using these as "reference" to justify the results of this specific method is questionable.

5. How do authors explain the loops found by CEASAR and not by Micro-C, if Micro-C is the gold standard? Can they specifically show that such loops are supported by Micro-C or Hi-C data still? Same goes for the many extra stripes they find.

6. At 1kb resolution, one could find many more loops than the 8k or so using a proper method instead of the reported "fast loop calling" approach the authors came up with. That raises the issue about this custom method as well as the comparison results showing the overlap among different methods. At least one other published way for loop calling and for stripe calling should be used to repeat all the results.

7. The authors show only two examples of CRISPRi which are known to be "functional". First of all, this is very limited and they could find many more examples in the literature, some from the papers I mentioned above. Second, even for the two loci, they are mentioning, they seem to pick only a small subset of validated connections. Could they recover the >1Mb loops in MYC locus for instance?

8. For Figure 5C, it is not clear how many loops these pile-ups are plotted with. For this analysis, it is critical to compare this enrichment with enrichment from a "control" cell line where the contact map is predicted by CEASAR (cross-comparison). That would also partially address my concern about the artificial visual patterns if they can show the enrichment patterns are specific (or at least more pronounced) for the cell type where the eQTLs are observed.

Minor

9. Figure S5 is missing

10. Showing what would happen to detected loops by omitting each of the epigenomic markers would be a better experiment to show the importance of each rather than just calculating their importance using "integrated gradient".

11. The 150 kb threshold for eQTLs is arbitrary. The authors claim their method performs best up until 200 kb. Why not at least use that threshold? Either way, this has to be explained and justified.

12. The justification of the use of a "surrogate" contact map from only a correlation curve is insufficient.

13. Line 158. "Appears" exclusively suggests visual inspection rather than quantification. This has to be clarified.

14. Line 180. It is really confusing to me how or why the peaks in the 1D features used are not informative. If this is the case then the reason behind has to be clarified.

Response to Reviewers

Thank you for reviewing our manuscript, “Connecting high-resolution 3D chromatin organization with epigenomics”. Below, we address each of the points raised by the two reviewers and describe the changes we have made to the manuscript. In what follows, the reviewer’s comments are given in black, interleaved with our responses in blue, and major changes of the main text in red.

Reviewer #1

The goal of this project is to use low-resolution Hi-C data and ChIP-Seq/ATAC-Seq to predict higher-resolution chromatin interaction maps. There have been at least 4-5 publications on this topic in different journals. Therefore, I think the novelty of this work is limited.

While we appreciate the reviewer’s feedback, we respectfully disagree with your statement about our work’s novelty. **Our work is novel in the following aspects.**

1. **CAESAR is the first deep-learning model connecting 3D genome organization with epigenomics at nucleosome resolution.** First, nucleosome resolution is the most appropriate resolution to connect epigenome and 3D chromatin organization because the majority of epigenomic changes (e.g., TF binding activities and histone modifications) reveal regulatory functions at nucleosome resolutions. By contrast, restriction fragments in Hi-C usually span over multiple nucleosomes. Therefore, it is more appropriate to build a predictive model (such as our CAESAR model) connecting epigenomics and Micro-C contact maps at nucleosome resolution. So far, our CAESAR is the only model built on Micro-C contact maps. Second, nucleosome resolution allows us to explore fine-scale regulatory patterns and relations (enhancer-promoter loops and eQTLs), which cannot be investigated on contact maps at 5 kb or lower resolutions yielded from previous methods including HiCPlus [1], HiCGAN [2], and HiC-Reg [3].
2. **CAESAR model is novel because it imputes all chromatin contacts and fine-scale structures** rather than only predicting specific sparse chromatin interactions like EpiTensor [4] does. Therefore, CAESAR provides the information of fine-scale structures like stripes, TADs, and polycomb interactions between repressive regions.
3. **Our work reaches an unprecedented scale of imputing 91 contact maps from human tissues, cell lines, primary cells, and *in vitro* differentiated cells.** None of the existing work has imputed chromatin contact maps so many human tissue and cell types.
4. **CAESAR is the first model among this category to include an attribution component, which provides detailed relationships between fine-scale structures and epigenomic features.** The previous work, such as HiCPlus [1] and HiCGAN [2], only used deep-learning models as “black boxes”, and did not interpret the model. With attribution, we were able to subtype fine-scale chromatin structures based on their relevant epigenomic features (see Supplementary Figure 6 in the revised manuscript).

Further, there is no evidence or functional validation to test the results from their algorithm.

In our study, we provided **both numerical evaluation and functional validation.**

Because Micro-C contact maps for hESC, HFF, and mESC are available, we performed **three sets of numerical evaluation experiments**, including cross-chromosome, cross-cell-type, and cross-species experiments, to directly test the predictive performance of our model. In the evaluation, the numerical evaluation metrics include

1. the correlation between our predicted contact maps and real Micro-C contact maps,
2. whether the predicted loops and stripes match with real Micro-C loops and stripes.

Note that in all of these cross-validation experiments, we separated the data into a training set, a tuning set and a hold-off test set. Numerical evaluations on the hold-off test set directly demonstrated that CAESAR is significantly more accurate than baselines. During the revision, we also added a new experiment in which we increased the contact distance range from 200 kb to 1 Mb. We observed that CAESAR still significantly outperformed baselines (Figure R2).

For the cell types without ground truth Micro-C contact maps, we performed **two indirect functional validation experiments**.

1. CAESAR predicts enhancer-promoter interactions detected by CRISPRi,
2. CAESAR predicts 3D interactions between eQTLs and their corresponding genes for 12 tissues and cell lines.

We believe that we have already provided enough evidence to justify the superior performance of the CAESAR model. Our lab is a pure computational lab and we currently do not have the wet lab facility to perform additional validation experiments.

My detailed comments are listed below:

Major:

1. Because a lot of downstream analysis in this work are based on the model trained with the surrogate Hi-C map, it would be crucial to add further validations. What are the accuracies of the loop and stripe predictions based on the model trained with surrogate Hi-C maps.

We followed your suggestion and added the accuracies of loop and stripe predictions based on the model trained with surrogate Hi-C maps. With this model, CAESAR captured 69% of the loops and 61% of the stripes on HFF. By contrast, CAESAR trained with matched Hi-C data captured 72% of the loops and 63% of the stripes on HFF. Therefore, both the loop/stripe accuracies and SCC measures demonstrate that surrogate contact maps can be used as an alternative if matched Hi-C contact maps are unavailable. We added this result to Supplementary Figure 3. It is noteworthy that the matched Hi-C and the surrogate Hi-C achieve very similar performance for fine-scale structure prediction, which is expected since the main contribution of Hi-C is to assist the model to relate genomic loci, and the information of fine-scale interactions mainly comes from epigenomic features. The functions of surrogate Hi-C contact maps are addressed comprehensively in our response to your point 4 below.

2. Resolutions used in this work are confusing. In the main text, they mentioned that they predicted the contact profile at 200 bp resolution (line #68). However, based on the supplementary notes, both loops and stripes were called at 1 kb.

Thank you for pointing this out. You are exactly correct that we did all imputation and analysis at 200 bp resolution, whereas loops and stripes were called at 1 kb. The reason is that **the most proper resolution for calling fine-scale loops and stripes is 1 kb**. In the original papers publishing mouse and human Micro-C contact maps [5, 6], their contact maps were also generated and visualized at 200 bp resolution, but loops were called at 1 kb or even lower resolutions (while stripes were not discussed in the original papers). Although fine-scale loops and stripes can only be detected in nucleosome-resolution Micro-C contact maps, these loops do have radii larger than 200 bp, and stripes have widths larger than 200 bp (an example region is provided in Figure R8 and the entire contact map is available at <https://data.4dnucleome.org/experiment-set-replicates/4DNESWST3UBH/>). The resolution and window size need to be comparable with loop radii and stripe widths so that the callers can detect the enriched signals effectively. We tested calling loops from the HFF Micro-C contact map with HICCUPS at 200 bp resolution with a 20-bin window, but we were only able to identify fewer than 10 loops after trying different sets of parameters.

3. It seems benchmark was not performed thoroughly. What is the performance of a model that directly predicts the entire contact map?

In original Micro-C papers[5, 6], both studies reported that Micro-C only outperforms Hi-C in the range of within 200 kb (Figure R1). This motivated us to only impute chromatin contacts within 200 kb. In addition, existing computational approaches also predict contacts within a certain distance, for example, 1~2 Mb for DeepHiC and HiCPlus, 1 Mb for HiC-Reg.

Figure R1: The comparison of contact frequencies between HFF Hi-C and Micro-C contact maps. Micro-C outperforms Hi-C in detecting contacts within 10^5 bp. In Micro-C contact maps, The majority of fine-scale loops also locate within 200 kb of genomic distance.

Nevertheless, we added a new experiment in which we extended CAESAR’s prediction range to 1 Mb (at a cost of reducing the resolution to 1 kb). In this range, CAESAR still outperforms baseline methods in terms of the stratum-adjusted correlation coefficient (SCC) with the observed Micro-C contact map (Figure R2).

Figure R2: Evaluation of CAESAR trained at 1kb resolution. **a**, An example in which CAESAR accurately predicts chromatin contacts within 1 Mb. **b**, The distance-stratified Pearson’s correlation with the observed Micro-C contact map from CAESAR and two baselines, HiC-Reg and HiCPlus, in a cross-chromosome experiment. **c**, The distance-stratified Pearson’s correlation with the observed Micro-C contact map from CAESAR in 1) a cross-chromosome experiment (train on hESC train set and test on hESC test set), 2) a cross-cell type experiment (train on HFF and test on hESC), and 3) a cross-species experiment (train on mESC and test on hESC).

4. According to Fig. 3c, using the surrogate Hi-C map can achieve a comparable performance to the model using cell-specific Hi-C map with 1 billion contacts. So, what is the contribution of Hi-C in the model?

In CAESAR, low-resolution Hi-C maps contributes to correctly aggregating epigenomic information from different genomic loci. Algorithmically, low-resolution Hi-C, represented as a graph in the graph convolutional networks, restricts feature aggregation in the neural network layers and guides the information flow between nodes.

We would like to use an example to illustrate. HiC-Reg, which does not use low-resolution Hi-C data, directly predicts the contact between two loci with the epigenomic features at the two loci as well as their distance. In Figure R3, we choose three ATAC-seq peaks **a**, **b**, and **c** which locate in two TADs. Since **b-c** distance is similar with **a-b**, without TAD information, the model is likely to predict contacts for both **a-b** and **b-c**. However, since **b** and **c** are in different TADs according to Hi-C contact maps, they are unlikely to strongly contact each other. When CAESAR predicts the contacts of locus **b**, with the help of Hi-C data, the model recognizes locus **a** as a closer neighbor, utilizes more information from **a** than from **c**, and is capable of making more accurate predictions.

Figure R3: An illustration of the contribution of low-resolution contact maps, in which **a**, **b**, and **c** are three loci, and **X** and **Y** are two contacts between them. From low-resolution Hi-C, we identify that **X** is an intra-TAD contact and **Y** is a cross-TAD contact.

We added an experiment in which no matched or surrogate Hi-C is used to train CAESAR. Instead, we generated a *fake contact map* as follows. First, the expected contact values at different distances were calculated from hESC and HFF Hi-C contact maps. Then each entry of the fake contact map was set to the expected contact value at the corresponding contact distance. Therefore, this fake Hi-C contact map does not have any informative structure. We observed that the performance of CAESAR with the fake Hi-C deteriorated dramatically (Figure R4). Therefore, a cell type-matched or surrogate Hi-C contact map is necessary for CAESAR.

Figure R4: Performance comparison between CAESAR trained with surrogate Hi-C and CAESAR trained with fake Hi-C. In terms of distance-stratified Pearson's correlation with the observed Micro-C contact map, CAESAR trained with surrogate Hi-C significantly outperforms CAESAR trained with fake Hi-C.

5. It has been shown that architectural stripes are frequently associated with active enhancers (H3K27ac). However, in Fig. 6 and Fig. S4, the authors showed that H3K27ac has negative attribution in predicting stripes. This is counter intuitive. Is this novel biological discovery or prediction error?

This is an excellent observation. **We identified additional evidence suggesting this is a novel discovery.** To illustrate this, we compare two well known active marks, H3K4me3 and H3K27ac in terms of their enrichment at stripe anchors (Figure R5).

Figure R5: The histograms of H3K4me3 and H3K27ac signal distribution in the genome *v.s* at stripe anchors. Most stripe anchors are highly enriched in H3K4me3, but less enriched in H3K27ac.

It is observed that most stripe anchors are highly enriched for H3K4me3. Among the 1,000 loci with the highest H3K4me3 signal on test set chromosomes, 374 are stripe anchors. By contrast, among the 1,000 loci with the highest H3K27ac signal on the same chromosomes, only 50 of them are stripe anchors. Therefore, although H3K4me3 and H3K27ac are both enriched in active regions, H3K4me3 shows a much higher enrichment at stripe anchors, and as a result, CAESAR connects stripes to positive H3K4me3 attribution. Instead, CAESAR is likely to regard H3K27ac as a feature related to “active regions but not stripes” and attributes negatively. Attribution reflects the correlation between fine-scale structures and epigenomic features learned by CAESAR, which is completely data-driven. We believe distinguishing these two epigenomic markers is out of the scope of our study, and deserves further investigation.

6. The APA plots throughout the paper look strange to me: the signals are enriched in the whole region (not sure if they used any normalization in these plots)?

The previous APA plots are based on OE-normalized contact maps. If you are concerned about the enrichment of signals in most regions, the reasons are as follows.

1. The window only includes 10 kb upstream and 10 kb downstream regions, so the contacts in the window are still likely to be associated with a gene or a regulatory element.
2. The majority of genes-eQTL pairs are in the active regions, in which chromatin contacts are more enriched.

Also in Fig. 5a-5b, it seems the loops and stripes are marked arbitrarily, because many other comparably evident dots are not marked.

Previously we only marked the loops which were relevant to our discussion. The other loops could also be called with a loop caller. We updated the figures (Figure 5a, Supplementary Figure 5e & f) in the revised manuscript, in which we labeled all the loops called by our loop caller.

Minor:

7. Fig. 5a-5b, Hi-C maps (the leftmost column) should use the same resolution to show the difference.

Thank you for your suggestion. We followed your advice and replaced the two figures (Supplementary Figure 5e & f in the revised manuscript) with 200 bp-resolution Hi-C contact maps. Since the original contact maps had never been processed into 200 bp resolution, we re-aligned the raw sequencing data from Schmitt et. al. [7] and processed them into 200 bp resolution. The nucleosome-resolution Hi-C contact maps are too sparse to show contact enrichment between eQTLs and genes (Figure R6).

Figure R6: The original human tissue Hi-C in Figure 5a and Figure 5b processed into 200 bp resolution. These Hi-C contact maps are too sparse to show contact enrichment between eQTLs and genes.

8. Not clear which map did the authors use to identify A/B compartments in fig. 3e, Hi-C, Micro-C, or the imputed map?

We used the A/B compartments called from Micro-C contact maps. The compartments are highly conserved between Hi-C and Micro-C contact maps (Figure R7). Since the imputed contact map does not have long-range interactions, it cannot be used to call compartments.

Figure R7: A/B compartments called from Hi-C (top) and Micro-C (bottom) visualized by IGV (HFF, chromosome 2) show high conservation.

9. In the method section (line 323), the authors mentioned that the pile-up analysis of the eQTL-TSS pairs was performed using Micro-C map of both hESC and HFF cell lines. However, I did not see the results for hESC Micro-C.

Thank you for pointing it out. We added the results for hESC Micro-C to our new Figure 5b in the revised manuscript.

Reviewer #2

The authors propose a supervised deep learning method that uses 1D chromatin features and low(er) resolution contact maps to predict high-resolution contact maps and their features including loops and stripes. This certainly could be a useful method however there are several important issues with the work and the manuscript as it stands.

Thank you for recognizing the significance of our work! Based on your suggestions, we updated our work in the following aspects.

- We resolved the problem of artificial stripe-like patterns in the imputed contact maps by adding positional encoding to the CAESAR model and reducing faraway contact weights during training.
- We cited additional work (including EpiTensor, DeepTACT, Akita, and DeepC) and added a detailed discussion regarding how CAESAR is innovatively different from existing work.

- We revised our loop caller by making it very close to a published loop caller HICCUPS. We explained the necessity of this change made to HICCUPS. Using a new bench-marking experiment, we justified our new loop caller is accurate and unbiased.
- We added a new experiment and validated that CAESAR accurately predicts enhancer-promoter interactions by predicting contacts near *MYC* region at longer contact distances and capturing interactions between additional four regulatory elements and *MYC*.

Major:

1. Unlike claimed, CEASAR is not “the first study connecting 3D genome organization with epigenomics”. Indeed, this journal has published in 2016 a method called EpiTensor that predicts Hi-C maps from 1D chromatin features. This work is not even cited! There are also several other methods such as DeepTACT (Bioinformatics, 2019), methods that predict contact maps from CTCF motifs and ChIP-seq data (the cited HiC-Reg method is just one of them) and two recent methods in Nature Methods that use DNA sequence to predict contacts (Fudenberg et al and Schwessinger et al). In short, the paper is missing in citations of critical literature making it seem more novel than it is. I fully understand the differences between the proposed method and all these other works and it is imperative to highlight these properly.

We cited additional work, including EpiTensor, DeepTACT, Akita, and DeepC, and tuned down our claim about our contribution to “CAESAR connects 3D genome organization with epigenomics at nucleosome resolution”. Comparing with the previous approaches, the novelties of CAESAR are as follows.

1. **CAESAR predicts 3D genome organization with epigenomics at nucleosome resolution.** First, nucleosome resolution is the most appropriate resolution to connect epigenome and 3D chromatin organization because the majority of epigenomic changes (TF binding activities and histone modifications) reveal regulatory functions at nucleosome resolutions. By contrast, restriction-fragments in Hi-C usually span over multiple nucleosomes. Therefore, it is more appropriate to build a predictive model (such as our CAESAR model) connecting epigenomics and Micro-C contact maps at nucleosome resolution. So far, our CAESAR is the only model built on Micro-C contact maps. Second, nucleosome resolution allows us to explore fine-scale regulatory patterns and relations (enhancer-promoter loops and eQTLs), which can be investigated on contact maps at 5 kb or lower resolutions yielded from previous methods including HiCPlus [1], HiCGAN [2], and HiC-Reg [3].
2. **CAESAR model imputes all chromatin contacts and fine-scale structures** rather than specific sparse chromatin interactions. Therefore, CAESAR captures more diverse fine-scale structures like stripes, TADs, and polycomb interactions between repressive regions. Previous methods EpiTensor [4] and DeepTACT [8] are different from CAESAR since they only predict sparse contacts between enhancers and promoters.
3. **CAESAR reaches an unprecedented scale of imputing 91 contact maps for human tissues, cell lines, primary cells, and *in vitro* differentiated cells.** None of the existing work has imputed so many human tissue and cell types. Akita [9] and DeepC [10] predict contact maps from DNA sequences. Since human cell lines and tissues have almost identical DNA sequences, they cannot generate cell line/tissue-specific contact maps like CAESAR.
4. **CAESAR is the first model among this category to include an attribution component, which provides detailed relationships between fine-scale structures and epigenomic features.** The previous work, such as HiCPlus [1] and HiCGAN [2], only used deep-learning models as “black boxes” and did not interpret the model or correlated input features to 3D structures. With attribution, we were able to subtype fine-scale chromatin structures based on their relevant epigenomic features (see Supplementary Figure 6 in the revised manuscript).

We updated the manuscript as follows.

(line 49 in the revised manuscript) CAESAR connects 3D genome organization with epigenomics at nucleosome resolution and unprecedented scale. First, compared with previous computational models for imputing Hi-C contact maps, such as HiCPlus [1], HiCGAN [2], and HiC-Reg [3], CAESAR reaches a much higher resolution. Since the majority of epigenomic activities (TF binding and histone modifications) take place at the nucleosome resolution, it is desirable to develop the predictive model that connects epigenomics and chromatin organization at the nucleosome resolution. Second, although previous models EpiTensor [4] and DeepTACT [8] also reconstruct sparse 3D chromatin interactions from epigenomics at an ultra-high resolution, CAESAR learns from real Micro-C contact maps and predicts all chromatin contacts within a distance range, which reveals diverse fine-scale structures such as stripes, TADs, and polycomb interactions between repressive regions. Third, CAESAR predicts contact maps from tissue-specific or cell line-specific epigenomic features instead of conserved DNA sequences. Therefore, it imputes an unprecedented number of high-resolution human chromatin contact maps, including 57 tissue samples, 16 cell lines, 12 primary cells, and 6 *in vitro* differentiated cells. The imputed high-resolution contact maps are shared on a web server (<https://nucleome.dcmf.med.umich.edu/>), which allows users to easily navigate these fine-scale chromatin structures and the corresponding explanatory epigenomic features. In addition, CAESAR is the first model among this category to include an attribution component, which reveals detailed relationships between 3D chromatin organization and epigenomic features.

2. In general, I find the predicted contact maps a bit artificial. The visual pattern (many stripe-like features) suggests that the method is highly biased by the 1D coverage of each region which gets reflected as the absence of depleted contact regions for some loci. This is immediately visible in Fig 1 Hi-C vs CEASAR vs Micro-C comparison. This issue will significantly impact the usefulness of the overall method for the broader community.

Thank you for the excellent observation. **This issue has been resolved after we made two changes to our CAESAR model.**

1. **A “positional encoding” module is added to the inputs to help the model identify the order of the genomic loci.** This is inspired by Transformer [11]. Graph neural networks only consider the edges between nodes (i.e., the Hi-C contacts between loci) but do not use the node order information (i.e., genomic coordinates). Therefore, in our previous model, the information of stripe anchor positions was extracted in the hidden representations, but to which nodes the stripe should span was learned less accurately. In this scenario, the model was likely to output a stripe that spans to every other node in the region. In our current model, the positional encoding provides the node order information, and helps CAESAR predict the end position of a stripe.
2. **We removed chromatin contacts whose value is below a threshold, and reduced the weights of longer-distance interactions during training.** The Micro-C contact maps are OE normalized (i.e., the average of each stratum is 1). At 180~200 kb contact distance, the contact is so sparse that there is only ~1 contact in every 1,000 bins, and therefore OE normalization amplifies these contact values (including some noise) by 1,000 times. As a result, the previous model predicted many false-positive long-range contacts. By thresholding contact values and adjusting their weights, CAESAR is less influenced by the amplified noise in these regions.

After the two changes, the artificial patterns such as long stripe-like patterns mostly disappeared (see new Figure 1b in our revised manuscript).

Figure 1b: In an example region, the polycomb interactions are accurately predicted by CAESAR. In another example region, loops and stripes undetected by Hi-C are accurately predicted by CAESAR.

3. The above issue is also apparent when comparing different entities in Figure 2c. The aggregate pattern of the CEASAR loops does not mimic that of Micro-C loops. There does not seem to be a distance-dependent decay as one moves away from the center pixel.

This issue has also been resolved in the new model. Figure 2 is also updated in our revised manuscript, and a distance-dependent decay of loops can be observed in the imputed contact map.

Figure 2d & 2f: **d**, The pile-up visualization of the loops called from 1) the input Hi-C contact map, 2) the CAESAR-imputed contact map, and 3) the observed Micro-C contact map. **f**, The pile-up visualization of the stripes called from 1) the input Hi-C contact map, 2) the CAESAR-imputed contact map, and 3) the observed Micro-C contact map.

4. The loop and stripe detection are done using simple and customized methods rather than established ways of doing these. Therefore, using these as “reference” to justify the results of this specific method is questionable.

For calling loops from our imputed high-resolution contact maps, we had to revise HICCUPS slightly for the following reason.

Currently published loop callers (e.g., HICCUPS and Mustache) do not directly apply to our imputed contact maps because they require a properly normalized contact map. Although HICCUPS documentation mentions the setting of “NONE” normalization, but executing the command “hiccups -cpu -k NONE” gives the error “Data not available”. CAESAR currently predicts chromatin contacts within a distance range (200 kb or 1 Mb) along the diagonal, which cannot be normalized by normalization methods (including KR, VC, and VCSQRT) which require the entire contact maps. **Therefore, we made a minor change to HICCUPS and implemented a new loop caller to replace our previous “fast loop calling” approach,** which searches for significantly enriched pixels with respect to the neighboring regions (see details in Supplementary Note 7 in the revised manuscript). The code of our loop caller has been made available on our GitHub repository (<https://github.com/liu-bioinfo-lab/caesar>).

We added a new experiment to compare original HICCUPS and our loop caller (revised HICCUPS) in detecting loops from real Micro-C contact maps (Figure R8a). We used “java -jar juicer_tools.jar hiccups -cpu -threads 0 -p 5 -i 10 -t 0.1 -f 0.1 -r 1000 -d 20000” to run HICCUPS. HICCUPS and our loop caller reported 13,308 loops and 20,089 loops respectively at 1 kb resolution from all chromosomes, in which 8,219 loops were called by both callers (Figure R8b). In a control experiment in which we called loops from two biological replicates of HFF Micro-C data, HICCUPS and our loop caller showed similar overlaps (Figure R8c). Therefore, the two callers are comparable, and our caller can be applied to CAESAR-imputed contact maps.

Figure R8: The comparison of HICCUPS and our loop caller. (a). The comparison of loops called by HICCUPS and our loop caller in two example regions. (b). The Venn diagram compares HICCUPS and our loop caller’s results from the HFF Micro-C contact map. (c). The overlap between reported loops on two replicates of HFF Micro-C is similar between HICCUPS and our loop caller.

For the stripe caller, the only paper which mentioned a stripe caller is Vian et al. [12]. However, they did not publish their tool “zebra” or provide the source code. Following their algorithm, we implemented “Quagga”, which searches for horizontal or vertical lines with contact signals significantly higher than the background (see details in Supplementary Note 8 in the revised manuscript). The code has been made available on our GitHub repository (<https://github.com/liu-bioinfo-lab/caesar>).

5. How do authors explain the loops found by CEASAR and not by Micro-C, if Micro-C is the gold standard? Can they specifically show that such loops are supported by Micro-C or Hi-C data still? Same goes for the many extra stripes they find.

All predictive models generate false positives. Following your suggestion, we carefully investigated the false positives produced by CAESAR. We observed that CAESAR’s false-positive loops and stripes fell into two categories.

1. The first category of false positives are supported by Micro-C data.
Due to the limited sequencing depth of Micro-C, some patterns are stripe-like or loop-like but not enriched enough for the callers to recognize. CAESAR enhances some of these structures to generate a false-positive but much clearer stripe (Figure R9a) or loop (Figure R9b). By raising the FDR threshold of callers from 0.10 to 0.20, 55% of false-positive stripes and 39% of false-positive loops can be called from the real Micro-C contact map.
2. The second category are not supported by Micro-C data. We manually went through many of these loops and stripes, and found two common patterns of false positives.
 - When there are a set of CTCF/ATAC-seq peaks in a small region without clear TAD separation, CAESAR may generate false-positive stripes on the peaks (Figure R9c) or loops between the peaks (Figure R9d).
 - “Isolated” CTCF and ATAC-seq peaks in repressed regions may result in false-positive stripes (Figure R9e) and loops (Figure R9f).

These false-positive patterns may indicate these “epigenomic-3D chromatin organization” patterns frequently exist in other genomic regions and have been learned by CAESAR. For example, the second region of Figure 1b in our revised manuscript is an example region whose pattern is quite similar to Figure R9d. Distinguishing between the two genomic regions may require additional epigenomic features or DNA sequence features.

Figure R9: Some examples for false-positive loops and stripes, in which **a**, **c**, and **e** are false-positive stripes, and **b**, **d**, and **f** are false-positive loops.

6. At 1kb resolution, one could find many more loops than the 8k or so using a proper method instead of the reported “fast loop calling” approach the authors came up with. That raises the issue about this custom method as well as the comparison results showing the overlap among different methods. At least one other published way for loop calling and for stripe calling should be used to repeat all the results.

The number of loops we previously reported was from the holdout test set (roughly 1/3 of the genome), which includes chr2, 5, 8, 11, 14, 15, 21, and 22. We expect the total number of loops within 200 kb to be ~20,000 whole-genome wide. We wonder whether this directly addressed your concern.

We also would like to point out that calling loops at different resolutions gives different sets of loops, and it is natural to have fewer Micro-C loops called at 1 kb resolution than at 5 kb resolution. For example, the loop highlighted with a green circle in Figure R8 can be called at 5 kb resolution but not at 1 kb resolution. In HFF Micro-C contact maps, 40,391 loops are called at 5 kb resolution with the default settings of HICCUPS, but only 14,937 loops are called at 1 kb resolution (with a 10 kb window and the default FDR). Among the 14,937 loops at 1 kb resolution, 13,308 (89.1%) have contact distance less than 200 kb (Figure R1), while only 57.4% of loops at 5 kb resolution do, which further demonstrates that predicting contacts within 200 kb provides sufficient fine-scale structural information.

The reasons why we did not use published loop caller or stripe caller on our imputed contact maps are provided in the response to your comment #4 above.

7. The authors show only two examples of CRISPRi which are known to be “functional”. First of all, this is very limited and they could find many more examples in the literature, some from the papers I mentioned above.

Thank you for the suggestion! We only evaluated CAESAR in two example regions because the number of CRISPRi enhancer screening datasets is quite limited. We tried to find additional published experiments, but the datasets we found, such as Canver et. al. and [13] and Smeenk et. al. [14], are from cell lines without sufficient epigenomic features for CAESAR to impute.

In addition, we tried to use the experiment used in Akita [9] and DeepC [10], but that example does not apply to CAESAR. In Akita [9] and DeepC [10], the authors use an example of genetically engineered deletion affecting 3D chromatin organization in HEK293T cell. Although this experiment is also based on CRISPR-Cas9 deletion, the goal is to directly evaluate the effect of CTCF site knockout on 3D chromatin structures. The edited cell lines do not have sufficient epigenomic features for CAESAR to impute its contact maps. By contrast, CRISPRi uses a group of single guide RNAs to target every possible site inside a region of interest, and cells expressing sgRNAs that target essential regulatory elements are depleted or increased in the final population. Therefore, CRISPRi is a tiling array to screen enhancer regions. Since we aim to demonstrate that 3D chromatin contacts predicted by CAESAR reveal enhancer-promoter interactions, CRISPRi enhancer screening is a more appropriate functional validation.

Second, even for the two loci, they are mentioning, they seem to pick only a small subset of validated connections. Could they recover the >1Mb loops in MYC locus for instance?

We increased the contact distance range to 1.25 Mb (at a cost of reducing the resolution to 1 kb), which allowed us to validate five additional regulatory elements within 1.25 Mb. Our imputed contact maps showed enriched contacts for four out of the five additional regulatory elements.

In Figure R10b, regulatory elements identified by CRISPRi are labeled with r1 and e1~e4 (e3 and e4 are only ~5 kb apart). The interaction between *MYC* and its “repressive element” r1 (PVT1) was already discussed in the paper. CAESAR predicts a stripe anchoring at *MYC* and connects *MYC* with a series of active and repressive regulatory elements. Along the stripe, *MYC*-enhancer interactions are also local maximum values.

Figure R10: **a**, Nine regulatory elements (seven active and two repressive) are revealed by CRISPRi data; **b**, The imputed 1 kb-resolution contact map of K562 shows the enriched contacts between *MYC* and four additional regulatory elements.

Another three distal enhancers, e5~e7, are located ~2 Mb from *MYC*, but we did not use them to validate our model for two reasons. First, Micro-C no longer outperforms Hi-C in this range, and Hi-C contact maps are already available for K562 (Figure R10a). Because CAESAR's superiority is to predict short-range interactions that Hi-C fails to detect (e.g., the *MYC*-r1 interaction), such an experiment will not help us validate our model. Second, it requires us to retrain the model to a larger range and this requires a lot of computational resources. If validating e5, e6 and e7 is critical to you, we are delighted to add this analysis in future revisions.

8. For Figure 5C, it is not clear how many loops these pile-ups are plotted with. For this analysis, it is critical to compare this enrichment with enrichment from a "control" cell line where the contact map is predicted by CAESAR (cross-comparison). That would also partially address my concern about the artificial visual patterns if they can show the enrichment patterns are specific (or at least more pronounced) for the

cell type where the eQTLs are observed.

Thank you for your suggestion. The number of eQTLs we used in this experiment has been added to Supplementary Table 5. Following your suggestion, we added the cross-comparison results of twelve tissues/cell types to Figure 5b in the revised manuscript. In the new figure, seven CAESAR-imputed contact maps (adrenal gland, heart left ventricle, IMR-90, pancreas, sigmoid colon, spleen, and transverse colon) have the highest contact values for their tissues/cell line-matched eQTL-TSS interactions. Another four CAESAR-imputed contact maps (GM12878, lung, stomach, and tibial nerve) also have close-to-highest contact values for their tissues/cell line-matched eQTL-TSS interactions. These results demonstrate that the tissue/cell type-specific enhancer-promoter interactions have been recovered by CAESAR.

Figure 5b: Pile-up analysis of the chromatin contacts between eQTLs and their corresponding gene TSS from twelve different human tissues and cell lines on CAESAR-imputed contact maps and Micro-C contact maps.

Minor:

9. Figure S5 is missing

Thank you for pointing that out. “Figure S5” in the previous supplementary notes should be “Figure S4”. This typo has been corrected in our revised manuscript.

10. Showing what would happen to detected loops by omitting each of the epigenomic markers would be a better experiment to show the importance of each rather than just calculating their importance using “integrated gradient”.

Thank you for your suggestion. We can think of two methods for “omitting an epigenomic marker to show the importance of features”. We are not sure which method you referred to, so we compare them with integrated gradient respectively.

In the first method, the model is not re-trained, and new outputs are generated by setting one input epigenomic feature to be all zeros. The loops/stripes in the new outputs are compared with the corresponding loop/stripe in the original outputs to evaluate the importance of the omitted epigenomic feature.

Integrated gradient calculates when a certain dimension of inputs increase from a background (often chosen to be zero) to the real value, how the outputs will change. Therefore, at most times, it is equivalent to omitting a dimension and observing the change of outputs. However, in some situations, integrated gradient is more desirable than simply omitting one dimension, especially in interpreting deep learning models with activation functions. Here is one simple example. One common activation function in neural networks is $Relu(a) = \max(0, a)$. Let $z = 1 - Relu(1 - x - y)$ be a simple network with inputs (x, y) and output z . Let $x = 1$ and $y = 1$, and therefore $z = 1$. By setting $x = 0$ (i.e., omitting x), $z = 1$ is not changed, but the conclusion that x is not important for predicting z is clearly wrong since omitting y gives exactly the same result. However, integrated gradient increases (x, y) from $(0, 0)$ to $(1, 1)$ and calculates the two dimensions’ contribution separately, which results in 0.5 attribution for both x and y .

In the second method, one epigenomic feature (e.g, H3K27ac) is omitted, and the model is re-trained with only five epigenomic features. After training, we can call loops/stripes from the predicted contact map and compare their strength with the previous model’s outputs. If the strength of a loop/stripe significantly changes, then the chosen epigenomic feature is important for predicting this loop/stripe.

However, we can only get an “overall” importance of the entire epigenomic feature of interest, but the outputs are not attributed to individual genomic locus. For example, integrated gradient describes which specific H3K4me3-enrich region contributes to CAESAR’s prediction of a stripe, but the second omitting approach only shows the feature H3K4me3 is important for predicting the stripe in a locus-agnostic fashion. To obtain the locus information, we need to omit epigenomic features from certain regions (i.e., certain dimensions in the input) instead of omitting the entire epigenomic track to train the model. However, there are ~ 90 million input dimensions in the genome (~ 3 billion base pairs \times 6 epigenomic features / 200 bp resolution) and it is impossible to omit every dimension to train different models.

11. The 150 kb threshold for eQTLs is arbitrary. The authors claim their method performs best up until 200 kb. Why not at least use that threshold? Either way, this has to be explained and justified.

Thank you for pointing this out. Since there is a 20 kb “window” in the pile-up analysis, we can only use eQTL-gene pairs within 180 kb (otherwise the window goes beyond 200 kb). In the revised manuscript, we changed it to 180 kb.

12. The justification of the use of a “surrogate” contact map from only a correlation curve is insufficient.

Thank you for your suggestion! **We added the accuracy of loop and stripe prediction based on the model trained with surrogate Hi-C maps as another evaluation metric.** With surrogate Hi-C contact maps, CAESAR still accurately predicts 69% of Micro-C loops and 61% of Micro-C stripes. By contrast, CAESAR trained with matched Hi-C data captured 72% of the loops and 63% of the stripes on HFF. Therefore, both the loop/stripe accuracies and SCC measures demonstrate that surrogate contact maps can be used as an alternative if matched Hi-C contact maps are unavailable. We added this result to Supplementary Figure 3.

It is noteworthy that the matched Hi-C and the surrogate Hi-C achieve very similar performance for fine-scale structure prediction, which is expected since the main contribution of Hi-C is to assist the model to relate genomic loci, and the information of fine-scale interactions mainly comes from epigenomic features. The functions of surrogate Hi-C contact maps are addressed comprehensively in the response to Reviewer #1 point 4.

13. Line 158. “Appears” exclusively suggests visual inspection rather than quantification. This has to be clarified.

Thank you for your suggestion. We called loops in these regions with the loop caller (FDR = 0.1) and marked all the called loops in the new Figure 5a and Supplementary Figure 5e & 5f in our revised manuscript.

14. Line 180. It is really confusing to me how or why the peaks in the 1D features used are not informative. If this is the case then the reason behind has to be clarified.

Attribution reflects the correlation between fine-scale structures and epigenomic features learned by CAESAR, which is completely data-driven. **Therefore, “not informative” features indicate CAESAR does not identify significant correlations between the features and the 3D structure.**

Supporting evidence can be found from raw data. We added an experiment to demonstrate why H3K27ac does not have high attribution in predicting stripes as another active feature H3K4me3. In Figure 6 and Figure S6, although H3K27ac is known to be an active mark and often overlaps with stripe anchors, by analyzing the original epigenomic signal, we confirmed that, unlike active mark H3K4me3, the most H3K27ac-enriched regions are less likely to overlap with stripe anchors. More precisely, among the 1,000 loci with the highest H3K4me3 signal on test set chromosomes, 374 are stripe anchors. By contrast, among the 1,000 loci with the highest H3K27ac signal on the same chromosomes, only 50 of them are stripe anchors. The distribution is also visualized with histograms in Figure R5 above. As a result, CAESAR connects stripes to positive H3K4me3 attribution but regards H3K27ac as a feature related to “active regions but not stripes” and attributes it negatively.

We have added the discussion to Supplementary Note 11 in our revised manuscript.

References

- [1] Y. Zhang, L. An, M. Hu, J. Tang, and F. Yue. HiCPlus: Resolution enhancement of Hi-C interaction heatmap. *bioRxiv*, April 2017. <https://doi.org/10.1101/112631>.
- [2] Qiao Liu, Hairong Lv, and Rui Jiang. hicGAN infers super resolution Hi-C data with generative adversarial networks. *Bioinformatics*, 35(14):i99–i107, 2019.
- [3] Shilu Zhang, Deborah Chasman, Sara Knaack, and Sushmita Roy. In silico prediction of high-resolution Hi-C interaction matrices. *Nature Communications*, 10(1):1–18, 2019.
- [4] Yun Zhu, Zhao Chen, Kai Zhang, Mengchi Wang, David Medovoy, John W Whitaker, Bo Ding, Nan Li, Lina Zheng, and Wei Wang. Constructing 3D interaction maps from 1D epigenomes. *Nature communications*, 7(1):1–11, 2016.
- [5] Tsung-Han S Hsieh, Claudia Cattoglio, Elena Slobodyanyuk, Anders S Hansen, Oliver J Rando, Robert Tjian, and Xavier Darzacq. Resolving the 3D landscape of transcription-linked mammalian chromatin folding. *Molecular Cell*, 2020.
- [6] Nils Krietenstein, Sameer Abraham, Sergey V Veney, Nezar Abdennur, Johan Gibcus, Tsung-Han S Hsieh, Krishna Mohan Parsi, Liyan Yang, René Maehr, Leonid A Mirny, et al. Ultrastructural details of mammalian chromosome architecture. *Molecular Cell*, 2020.

- [7] A. D. Schmitt, M. Hu, I. Jung, Z. Xu, Y. Qiu, C. L. Tan, Y. Li, S. Lin, Y. Lin, C. L. Barr, and B. Ren. A compendium of chromatin contact maps reveals spatially active regions in the human genome. *Cell Reports*, 17:2042–2059, 2016.
- [8] Wenran Li, Wing Hung Wong, and Rui Jiang. DeepTACT: predicting 3D chromatin contacts via bootstrapping deep learning. *Nucleic acids research*, 47(10):e60–e60, 2019.
- [9] Geoff Fudenberg, David R Kelley, and Katherine S Pollard. Predicting 3D genome folding from DNA sequence with akita. *Nature Methods*, 17(11):1111–1117, 2020.
- [10] Ron Schwessinger, Matthew Gosden, Damien Downes, Richard C Brown, A Marieke Oudelaar, Jelena Telenius, Yee Whye Teh, Gerton Lunter, and Jim R Hughes. DeepC: predicting 3D genome folding using megabase-scale transfer learning. *Nature Methods*, 17(11):1118–1124, 2020.
- [11] Ashish Vaswani, Noam Shazeer, Niki Parmar, Jakob Uszkoreit, Llion Jones, Aidan N Gomez, Lukasz Kaiser, and Illia Polosukhin. Attention is all you need. *arXiv preprint arXiv:1706.03762*, 2017.
- [12] Laura Vian, Aleksandra Pekowska, Suhas SP Rao, Kyong-Rim Kieffer-Kwon, Seolkyoung Jung, Laura Baranello, Su-Chen Huang, Laila El Khattabi, Marei Dose, Nathanael Pruett, et al. The energetics and physiological impact of cohesin extrusion. *Cell*, 173(5):1165–1178, 2018.
- [13] Matthew C Canver, Elenoe C Smith, Falak Sher, Luca Pinello, Neville E Sanjana, Ophir Shalem, Diane D Chen, Patrick G Schupp, Divya S Vinjamur, Sara P Garcia, et al. BCL11A enhancer dissection by Cas9-mediated in situ saturating mutagenesis. *Nature*, 527(7577):192–197, 2015.
- [14] Leonie Smeenk, Sophie Ottema, Roger Mulet-Lazaro, Anja Ebert, Marije Havermans, Andrea Arricibita Varea, Michaela Fellner, Dorien Pastoors, Stanley van Herk, Claudia Erpelinck-Verschueren, et al. Selective requirement of MYB for oncogenic hyperactivation of a translocated enhancer in leukemia. *Cancer Discovery*, 2021.

REVIEWER COMMENTS

Reviewer #1 (Remarks to the Author):

Overall, the paper improved a lot during revision.

My remaining concern is that our group has not been able to run their software. The GitHub for CAESAR didn't provide any of the models they've trained. Also, there is no direct packages for training and prediction, which means that the users need to revise their source code for training and prediction, which is not trivial.

Reviewer #3 (Remarks to the Author):

In this manuscript, Feng et al. develop a supervised deep learning algorithm, CAESAR, to predict nucleosome-resolution contact maps from lower resolution Hi-C and epigenomic tracks. Compared to previously published methods performing similar tasks, I can see two advantages of CAESAR that would make it valuable for the community: the increase in resolution and, most of all, the ability to predict cell type specific contact maps. My primary concern however is that this second claim is not sufficiently demonstrated in the current manuscript. Besides that, I think the study is sound and the authors have addressed previous reviewers' technical concerns.

Below are my specific comments.

Major comments:

1) The ability to predict cell type specific contact maps should be more thoroughly demonstrated, as it is one main novelty of the method. To this aim the authors could apply similar metrics as those already used in their manuscript. First, they could measure Spearman's correlations between pairs of predicted contact maps in different cell types. Then, they should compare the obtained values with correlations between experimental maps (micro-C and/or Hi-C) in different cell types to check if the overall level of variability is similar. As a control, they could use correlations between experimental replicates in the same cell type. I think this gives already a good indication of the overall method performance in distinguishing cell types. Second, to evaluate the ability to capture specific, differential contact patterns, they could measure, comprehensively, the model accuracy in distinguishing cell type specific structures such as loops and stripes.

2) CAESAR is benchmarked against two published methods and shown to outperform them. However, more recent methods published in Nature Methods 2020, Fudenberg, et al. (Akita), and Schwessinger, et al. (DeepC) are not even cited. At least a brief discussion about these methods and, if applicable, some quantitative comparisons should be added.

3) In particular, Akita is also trained on micro-C data, so CAESAR is not the first to do, as they seem to suggest.

4) In Fig. 3c it is shown that if surrogate Hi-C is used in place of the original Hi-C, the model still gives good performance in terms of SCC. I wonder if it could be simply used Hi-C from a different cell type instead of surrogate Hi-C. Does surrogate Hi-C perform better than Hi-C from another cell type?

5) In Fig. 4a, I can't see any CRISPRi score peaks at the PVT1 region.

6) Are the predicted contact data provided in the web server at 200bp resolution? Indeed, when I try to extract the data with JuiceTools dump, I get an error message saying the data are only available at 25000 10000 5000 1000 resolutions. Please clarify, and sorry if this is just my mistake.

Minor:

7) What is the source of Hi-C data in the pancreas shown in Fig.5a? There is no reference in the maintext and in Suppl. Table 2, where Hi-C data sources are listed, there is no pancreas Hi-C. All employed datasets should be referenced.

8) In the GitHub repository, unlike written in the Utiles/README file, no example codes are provided for reproducing downstream analysis shown in the paper figures 2-6. Additionally, there is no README for the usage of the developed loop and stripe callers.

Response to Reviewers

Thank you for reviewing our manuscript titled “Connecting high-resolution 3D chromatin organization with epigenomics”. Below, we address each of the points raised by the two reviewers and describe the changes we have made to the manuscript. In what follows, the reviewer’s comments are given in black, interleaved with our responses in blue, and major changes of the main text in red.

Reviewer #1

Overall, the paper improved a lot during revision.

My remaining concern is that our group has not been able to run their software. The GitHub for CAESAR didn’t provide any of the models they’ve trained. Also, there is no direct packages for training and prediction, which means that the users need to revise their source code for training and prediction, which is not trivial.

Thank you for your advice!

In our previous version, we had published our models in .h5 files on GitHub (under the “/Model” directory). We apologize that we did not make them obvious.

In our latest version, we re-organized the software code and provided two options for users to use our model and code easily. The first option is for users who would like to directly use the trained CAESAR to impute their own contact maps. In this option, the users can directly use the trained model provided in .h5 files (under the “/Imputation” directory). The second option is for users who would like to train the model (which requires large storage and memory) from scratch by themselves. In this option, the users can follow our step-by-step instructions about data processing and model training procedure, which are also shared on our GitHub repository (under the “/Training” directory).

Reviewer #3

In this manuscript, Feng et al. develop a supervised deep learning algorithm, CAESAR, to predict nucleosome-resolution contact maps from lower resolution Hi-C and epigenomic tracks. Compared to previously published methods performing similar tasks, I can see two advantages of CAESAR that would make it valuable for the community: the increase in resolution and, most of all, the ability to predict cell type specific contact maps. My primary concern however is that this second claim is not sufficiently demonstrated in the current manuscript. Besides that, I think the study is sound and the authors have addressed previous reviewers’ technical concerns.

Thank you for recognizing the advantages of our work. Following your suggestions, we improved our work in the following ways.

- We cited additional work (including Akita and DeepC) and discussed how CAESAR is different from existing work.
- We found that the variability between imputed cell-type-specific contact maps is comparable with that between experimental H1 and HFF Micro-C contact maps, which suggested that CAESAR can predict cell type-specific contact maps.
- We added new experiments suggesting that some of the fine-scale structures including loops and stripes predicted by CAESAR are cell type-specific.
- We added new experiments showing that CAESAR trained with cell type-mismatched Hi-C only moderately underperformed CAESAR trained with surrogate Hi-C.
- We modified Fig. 4a to better visualize the regulatory elements uncovered by CRISPRi scores.
- We re-organized our code on the GitHub repository and provided a trained model in .h5 format as well as the code for generating the results in our paper.

Below are my specific comments.

Major:

1. The ability to predict cell type specific contact maps should be more thoroughly demonstrated, as it is one main novelty of the method. To this aim the authors could apply similar metrics as those already used in their manuscript. First, they could measure Spearman's correlations between pairs of predicted contact maps in different cell types. Then, they should compare the obtained values with correlations between experimental maps (micro-C and/or Hi-C) in different cell types to check if the overall level of variability is similar. As a control, they could use correlations between experimental replicates in the same cell type. I think this gives already a good indication of the overall method performance in distinguishing cell types.

Thank you for your suggestions.

We added one experiment to show the ability to predict cell-type-specific contact maps. In total, our study imputed high-resolution chromatin contact maps for sixteen human cell lines. Following your suggestion, we calculated the similarity between imputed contact maps with HiCRep [1]. The pairwise similarity matrix between the sixteen cell lines is visualized in Fig. R1. As a control, we also calculated the HiCRep scores between experimental Micro-C contact maps from biological replicates of H1 and HFF.

The HiCRep scores between the sixteen cell lines (mean=0.77, std=0.04) are comparable with the scores between H1 and HFF (mean=0.74, std=0.03), whereas HiCRep scores between two biological replicates are as high as 0.85 (H1) and 0.87 (HFF). Therefore, CAESAR-imputed contact maps of human cell lines show similar variability as experimental Micro-C contact maps, indicating CAESAR's capability of distinguishing cell types. We have included this result in Supplementary Figure S5 in our revised manuscript.

Figure R1: The heatmap of HiCRep reproducibility scores between Micro-C contact maps from HFF/H1 biological replicates (above) and the heatmap of HiCRep reproducibility scores between the imputed contact maps of the sixteen cell lines used in our study (below).

Second, to evaluate the ability to capture specific, differential contact patterns, they could measure, comprehensively, the model accuracy in distinguishing cell type specific structures such as loops and stripes.

Figure R2: CAESAR predicts cell type-specific fine-scale structures. (a and c) Sixteen sets of loops and stripes were called from CAESAR-imputed contact maps of sixteen cell lines, and the APA analysis of these loop/stripe regions was carried out across the sixteen cell lines' imputed contact maps. (b and d) Contacts are significantly more enriched in the APA analysis when the set of loops/stripes and the contact map are from the matched cell line.

Since we do not have the ground truth fine-scale structures in Micro-C contact maps from these cell lines, we instead added another experiment to illustrate that some of the loops and stripes predicted by CAESAR are cell type-specific.

From the imputed high-resolution chromatin contact maps for the sixteen human cell lines, we first called loops and stripes using the loop and stripe callers described in our manuscript,

and achieved sixteen sets of loops and stripes. Aggregated peak analysis (APA) was then carried out across the sixteen human cell lines, in which we piled up the contacts in the called loop and stripe regions, and calculated contact enrichment scores for these sixteen sets of loops and stripes in the sixteen sets of contact maps. The APA analysis produced a 16-by-16 matrix with rows corresponding to the sixteen sets of loops and stripes, and columns corresponding to the sixteen contact maps in which we calculated the contact enrichment scores (Fig. R2a and R2c). Therefore in this 16-by-16 matrix, diagonal elements are cell-type-matched whereas off-diagonal elements are cell-type-mismatched. Under the null hypothesis that these loops and stripes are not cell-type-specific, the enrichment scores from diagonal elements and those from off-diagonal elements should be similarly distributed. It is observed that chromatin contacts in loop/stripe regions called from one cell type are significantly more enriched in the cell-type-matched contact map predicted by CAESAR (Fig. R2b and R2d). This demonstrates that some fine-scale structures predicted by CAESAR are cell type-specific. We have included this result in Supplementary Figure S6 in our revised manuscript.

(Line 119) Chromatin contact maps imputed by CAESAR also show comparable cell-type variability as real Micro-C contact maps in terms of SCC and cell type-specific fine-scale structures including chromatin loops and stripes (Figures S5 and S6; Supplementary Note 10).

2. CAESAR is benchmarked against two published methods and shown to outperform them. However, more recent methods published in Nature Methods 2020, Fudenberg, et al. (Akita), and Schwessinger, et al. (DeepC) are not even cited. At least a brief discussion about these methods and, if applicable, some quantitative comparisons should be added.

Thank you for the advice. We cited Akita and DeepC and added a brief discussion in our revised manuscript.

(Line 49) CAESAR connects 3D genome organization with epigenomics at nucleosome resolution and unprecedented scale. First, compared with previous computational models for imputing Hi-C contact maps, such as HiCPlus [2], HiCGAN [3], and HiC-Reg [4], CAESAR reaches a much higher resolution. Since the majority of epigenomic activities (TF binding and histone modifications) take place at the nucleosome resolution, it is desirable to develop the predictive model that connects epigenomics and chromatin organization at the nucleosome resolution. Second, although previous models EpiTensor [5] and DeepTACT [6] also reconstruct sparse 3D chromatin interactions from epigenomics at an ultra-high resolution, CAESAR learns from real Micro-C contact maps and predicts all chromatin contacts within a distance range, which reveals diverse fine-scale structures such as stripes, TADs, and polycomb interactions between repressive regions. Third, different from Akita [7] and DeepC [8] which predict chromatin contact maps from conserved DNA sequences, CAESAR generates tissue-specific or cell line-specific predictions from epigenomic features. Therefore, it imputes an unprecedented number of high-resolution human chromatin contact maps, including 57 tissue samples, 16 cell lines, 12 primary cells, and 6 *in vitro* differentiated cells. The imputed high-resolution contact maps are shared on a web server (<https://nucleome.dcmf.med.umich.edu/>), which allows users to easily navigate these fine-scale chromatin structures and the corresponding explanatory epigenomic features. In addition, CAESAR includes an attribution component, which reveals detailed relationships between 3D chromatin organization and epigenomic features.

It is difficult for us to directly compare the performance between CAESAR, Akita and DeepC due to the different resolutions they used, and the evaluation metric SCC is quite sensitive to resolution. Since the SCC value between the same two contact maps is higher at a lower resolution, it is a fair comparison only if imputed contact maps have exactly the same resolution.

According to Fig. 1d of the DeepC paper [8], they reached ~ 0.6 SCC at 5 Kb resolution in the range within 200 kb, while CAESAR achieved ~ 0.8 (Fig. 2a in our manuscript) at 200 bp resolution. In addition, CAESAR is more efficient since it requires fewer parameters than DeepC (2 million v.s. 60 million).

In Akita, the 2-fold pooling operation in convolution layers results in the 2^N bp resolution ($N = 11$ and resolution = 2048 bp in their original paper), which does not match CAESAR’s resolution for a fair comparison as well. Nevertheless, we followed Akita’s strategy and published the predicted contact maps of chromosome 17 on our web server.

3. In particular, Akita is also trained on micro-C data, so CAESAR is not the first to do, as they seem to suggest.

We have tuned down our claim to “(Line 49) CAESAR connects 3D genome organization with epigenomics at nucleosome resolution and unprecedented scale”.

4. In Fig. 3c it is shown that if surrogate Hi-C is used in place of the original Hi-C, the model still gives good performance in terms of SCC. I wonder if it could be simply used Hi-C from a different cell type instead of surrogate Hi-C. Does surrogate Hi-C perform better than Hi-C from another cell type?

We added two experiments in which CAESAR was trained with K562 and GM12878 and predicted high-resolution contact maps of HFF. In terms of distance-stratified Pearson’s correlation with the observed Micro-C contact map, CAESAR trained with surrogate Hi-C slightly outperforms CAESAR trained with mismatched Hi-C (GM12878 or K562) (Fig. R3). However, CAESAR models trained with deep-sequenced GM12878 and K562 Hi-C also reached much higher correlation than the original HFF Hi-C, indicating that these models still predict Micro-C contacts well.

Figure R3: The performance of CAESAR trained with the cell-type-matched Hi-C (HFF), a surrogate Hi-C aggregated from 5 cell lines, and Hi-C from two other cell types evaluated by SCC with observed HFF Micro-C contact map. The black dotted line shows the correlation between HFF Hi-C and HFF Micro-C contact maps.

5. In Fig. 4a, I can’t see any CRISPRi score peaks at the PVT1 region.

Thank you for the question! The data were from Fig. 2a of Fulco et. al.[9] (Fig. R4), in which the corresponding peak was marked as r1 by original authors.

The peak we referred to is negative and less significant than the positive ones. Therefore, only visualizing the tracks with proper resolutions and scales can avoid a misleading visual effect. We now changed the resolution to 200 bp and adjusted the scale of the y-axis to make the peak more obvious (Fig. R5).

Figure R4: CRISPRi screening identifies seven distal enhancers (e1 to e7) that activate MYC and two repressive elements (r1, r2) that may act to repress MYC.

Figure R5: The CRISPRi scores of K562 near MYC region.

6. Are the predicted contact data provided in the web server at 200bp resolution? Indeed, when I try to extract the data with JuiceTools dump, I get an error message saying the data are only available at 25000 10000 5000 1000 resolutions. Please clarify, and sorry if this is just my mistake.

You are correct that the contact maps published on our server only have 25 K, 10 K, 5 K, and 1 K resolutions. We did not publish whole-genome contact maps at 200 bp resolution due to storage concerns. Imputed contact maps at 200 bp resolution will take over ~ 4 TB of storage on our web server.

Since it is easier to re-generate than download the contact maps for the whole genome, we provide the code for generating contact maps of any user-specified regions in our GitHub repository. We also agree that 200-bp-resolution contact maps are significant results of CAESAR, and therefore, we followed how Akita [7] shared the predicted contact map and published the 200-bp contact maps of chromosome 17 for all human cell lines and tissue types.

Minor:

7. What is the source of Hi-C data in the pancreas shown in Fig.5a? There is no reference in the main text and in Suppl. Table 2, where Hi-C data sources are listed, there is no pancreas Hi-C. All employed datasets should be referenced.

The human tissue Hi-C contact maps including heart left ventricle, lung, and pancreas are all from Schmitt et al.[10]. We have added the data source to Supplementary Table 2.

8. In the GitHub repository, unlike written in the Utiles/README file, no example codes are provided for reproducing downstream analysis shown in the paper figures 2-6. Additionally, there is no README for the usage of the developed loop and stripe callers.

We re-organized our GitHub repository and added additional information to the README file including how to impute a customized region with CAESAR (under the “/Imputation” directory), how to train CAESAR from scratch (under the “/Training” directory), and how to run the loop caller and stripe caller (under the “/Manuscript” directory). Generating the figures in the manuscript requires large raw data which are stored locally, but the code for generating these data is also provided.

References

- [1] T. Yang, F. Zhang, G. G. Yardımcı, F. Song, R. C. Hardison, W. S. Noble, F. Yue, and Q. Li. HiCRep: assessing the reproducibility of Hi-C data using a stratum-adjusted correlation coefficient. *Genome Research*, 27(11):1939–1949, 2017.
- [2] Y. Zhang, L. An, M. Hu, J. Tang, and F. Yue. HiCPlus: Resolution enhancement of Hi-C interaction heatmap. *bioRxiv*, April 2017. <https://doi.org/10.1101/112631>.
- [3] Qiao Liu, Hairong Lv, and Rui Jiang. hicGAN infers super resolution Hi-C data with generative adversarial networks. *Bioinformatics*, 35(14):i99–i107, 2019.
- [4] Shilu Zhang, Deborah Chasman, Sara Knaack, and Sushmita Roy. In silico prediction of high-resolution Hi-C interaction matrices. *Nature Communications*, 10(1):1–18, 2019.
- [5] Yun Zhu, Zhao Chen, Kai Zhang, Mengchi Wang, David Medovoy, John W Whitaker, Bo Ding, Nan Li, Lina Zheng, and Wei Wang. Constructing 3D interaction maps from 1D epigenomes. *Nature communications*, 7(1):1–11, 2016.
- [6] Wenran Li, Wing Hung Wong, and Rui Jiang. DeepTACT: predicting 3D chromatin contacts via bootstrapping deep learning. *Nucleic acids research*, 47(10):e60–e60, 2019.
- [7] Geoff Fudenberg, David R Kelley, and Katherine S Pollard. Predicting 3D genome folding from DNA sequence with akita. *Nature Methods*, 17(11):1111–1117, 2020.
- [8] Ron Schwessinger, Matthew Gosden, Damien Downes, Richard C Brown, A Marieke Oudelaar, Jelena Telenius, Yee Whye Teh, Gerton Lunter, and Jim R Hughes. DeepC: predicting 3D genome folding using megabase-scale transfer learning. *Nature Methods*, 17(11):1118–1124, 2020.
- [9] Charles P Fulco, Mathias Munschauer, Rockwell Anyoha, Glen Munson, Sharon R Grossman, Elizabeth M Perez, Michael Kane, Brian Cleary, Eric S Lander, and Jesse M Engreitz. Systematic mapping of functional enhancer–promoter connections with CRISPR interference. *Science*, 354(6313):769–773, 2016.
- [10] A. D. Schmitt, M. Hu, I. Jung, Z. Xu, Y. Qiu, C. L. Tan, Y. Li, S. Lin, Y. Lin, C. L. Barr, and B. Ren. A compendium of chromatin contact maps reveals spatially active regions in the human genome. *Cell Reports*, 17:2042–2059, 2016.

REVIEWERS' COMMENTS

Reviewer #3 (Remarks to the Author):

The authors have addressed my concerns in the revisions, and I think the manuscript is suitable for publication.